# A GEOMETRIC PERSPECTIVE ON DIFFUSION MODELS

## ABSTRACT

Recent years have witnessed significant progress in developing effective training and fast sampling techniques for diffusion models. A remarkable advancement is the use of stochastic differential equations (SDEs) and their marginal-preserving ordinary differential equations (ODEs) to describe data perturbation and generative modeling in a unified framework. In this paper, we carefully inspect the ODE-based sampling of a popular variance-exploding SDE and reveal several intriguing structures of its sampling dynamics. We discover that the data distribution and the noise distribution are smoothly connected with a quasi-linear *sampling trajectory* and another implicit *denoising trajectory* that even converges faster. Meanwhile, the denoising trajectory governs the curvature of the corresponding sampling trajectory and its various finite differences yield all second-order samplers used in practice. Furthermore, we establish a theoretical relationship between the optimal ODE-based sampling and the classic mean-shift (mode-seeking) algorithm, with which we can characterize the asymptotic behavior of diffusion models and identify the empirical score deviation.

## 1 INTRODUCTION

Diffusion models, or score-based generative models (Sohl-Dickstein et al., 2015; Song & Ermon, 2019; Ho et al., 2020; Song et al., 2021c) have attracted growing attention and seen impressive success in various domains, including image (Dhariwal & Nichol, 2021; Rombach et al., 2022), video (Ho et al., 2022; Blattmann et al., 2023), audio (Kong et al., 2021; Chen et al., 2021), and especially text-to-image synthesis (Saharia et al., 2022; Ruiz et al., 2023). Such models are essentially governed by a certain kind of stochastic differential equations (SDEs) that smooth data into noise in a forward process and then generate data from noise in a backward process (Song et al., 2021c).

Generally, the probability density in the forward SDE evolves through a spectrum of Gaussian *kernel density estimates* of the original data with varying bandwidths. As such, one can couple theoretically infinite data-noise pairs and train a noise-dependent neural network (*a.k.a.* diffusion model) to minimize the mean square error for data reconstruction. Once such a denoising model with sufficient capacity is well optimized, it will faithfully capture the score (gradient of the log-density *w.r.t.* the input) of the data density smoothed with various levels of noise (Raphan & Simoncelli, 2011; Bengio et al., 2013; Karras et al., 2022). The generative ability is then emerged by simulating the (score-based) backward SDE with any numerical solvers. Alternatively, we can simulate the corresponding ordinary differential equation (ODE) that preserves the same marginal distributions as the SDE (Song et al., 2021c;a; Lu et al., 2022; Zhang & Chen, 2023). The deterministic ODE-based sampling gets rid of the stochasticity, apart from the randomness of drawing initial samples, and thus makes the whole generative process more comprehensible and controllable (Song et al., 2021a; Karras et al., 2022). However, more details about how diffusion models behave under this dense mathematical framework are still currently unknown.

In this paper, we provide a geometric perspective to facilitate an intuitive understanding of diffusion models, especially their sampling dynamics. The state-of-the-art variance-exploding SDE (Karras et al., 2022) is taken as the main example to reveal the underlying intriguing structures. Our empirical observations (Section 3) are summarized and illustrated in Figure 1. Given an initial sample from the noise distribution, the difference between its denoising output and its current position forms the score direction for simulating the sampling trajectory. This explicit trajectory is almost straight such that the ODE simulation can be greatly accelerated at a modest cost of truncation error. Besides, the denoising output itself forms another implicit trajectory that quickly appears decent visual quality,

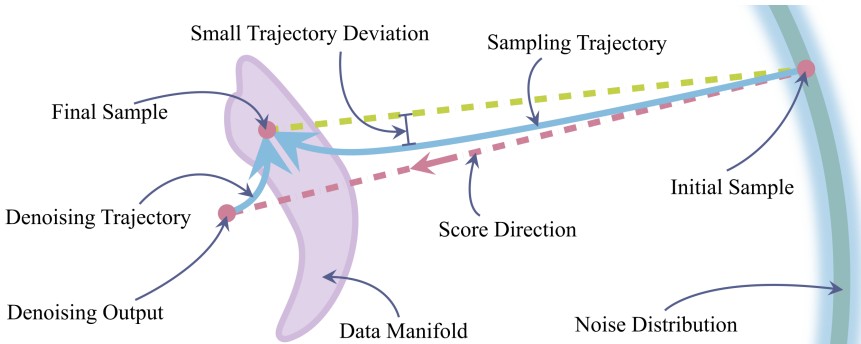

Figure 1: The geometric picture of ODE-based sampling in diffusion models. An initial sample (from the noise distribution) starts from a big sphere and converges to its final sample (in the data manifold) along a smooth, quasi-linear sampling trajectory. Meanwhile, its denoising output lays in an implicit, smooth denoising trajectory starting from the approximate dataset mean. The denoising output is relatively close to the final sample and converges much faster in terms of visual quality.

which offers a simple way to accelerate existing samplers (Section 3.2). Intriguingly, the derivative of denoising trajectory shares the same direction as the negative second-order derivative of sampling trajectory, and in principle, all previously developed second-order samplers can be derived from the specific finite differences of the denoising trajectory (Section 4). Overall, these two trajectories fully depict the ODE-based sampling process in diffusion models.

Furthermore, we establish a theoretical relationship between the optimal ODE-based sampling and (annealed) mean shift (Comaniciu & Meer, 2002; Shen et al., 2005), which implies that each single Euler step in sampling actually moves the given sample to a convex combination of annealed mean shift and its current position. Meanwhile, the sample likelihood increases from the current position to the vicinity of the mean-shift position. This property guarantees that under a mild condition, the likelihood of each sample in the denoising trajectory consistently surpasses its counterpart from the sampling trajectory (Section 5), and thus the visual quality of the former generally exceeds that of the latter. The theoretical connection also helps to identify different behaviors of the empirical score, and based on which, we argue that a slight score deviation from the optimum ensures the generative ability of diffusion models while greatly alleviating the mode collapse issue (Section 6). Finally, the geometric perspective enables us to better understand distillation-based consistency models (Song et al., 2023) (Appendix D) and latent interpolations in practice (Appendix E).

## 2 PRELIMINARIES

We begin with a brief overview of the basic concepts in developing score-based generative models. With the tool of *stochastic differential equations* (SDEs), the data perturbation in diffusion models is modeled as a continuous stochastic process $\{\mathbf{x}_t\}_{t=0}^T$ (Song et al., 2021c; Karras et al., 2022):

$$\mathrm{d}\mathbf{x} = \mathbf{f}(\mathbf{x}, t)\mathrm{d}t + g(t)\mathrm{d}\mathbf{w}_t, \qquad \mathbf{f}(\cdot, t) : \mathbb{R}^d \to \mathbb{R}^d, \quad g(\cdot) : \mathbb{R} \to \mathbb{R}, \tag{1}$$

where $\mathbf{w}_t$ is the standard Wiener process; $\mathbf{f}(\cdot, t)$ and $g(t)$ are drift and diffusion coefficients, respectively (Oksendal, 2013). We denote the distribution of $\mathbf{x}_t$ as $p_t(\mathbf{x})$ and such an Itô SDE smoothly transforms the empirical data distribution $p_0(\mathbf{x}) = p_d(\mathbf{x})$ to the (approximate) noise distribution $p_T(\mathbf{x}) \approx p_n(\mathbf{x})$ in a forward manner. By properly setting the coefficients, some established models referred to as variance-preserving (VP) and variance-exploding (VE) SDEs can be recovered (Song & Ermon, 2019; Ho et al., 2020; Song et al., 2021c). The reversal of Eq. (1) is another SDE that allows to synthesize data from noise in a backward manner (Feller, 1949; Anderson, 1982). Remarkably, there exists a *probability flow ordinary differential equation* (PF-ODE) sharing the same marginal distribution $\{p_t(\mathbf{x})\}_{t=0}^T$ as the reverse SDE at each time step of the diffusion process:

$$\mathrm{d}\mathbf{x} = \left[ \mathbf{f}(\mathbf{x}, t) - \frac{1}{2}g(t)^2 \nabla_\mathbf{x} \log p_t(\mathbf{x}) \right] \mathrm{d}t. \tag{2}$$

The deterministic nature of ODE offers several benefits including efficient sampling, unique encoding, and meaningful latent manipulations (Song et al., 2021c;a). We thus choose Eq. (2) to ana-

lyze model behaviors throughout this paper. Simulating the above ODE requests having the *score function* $\nabla_{\mathbf{x}} \log p_t(\mathbf{x})$ in hand (Hyvärinen, 2005; Lyu, 2009), which is typically estimated with the *denoising score matching* (DSM) criterion (Vincent, 2011; Song & Ermon, 2019). From the perspective of *empirical Bayes* (Robbins, 1956; Efron, 2011; Saremi & Hyvärinen, 2019), there exists a profound connection between DSM and *denoising autoencoders* (DAEs) (Vincent et al., 2008; Bengio et al., 2013; Alain & Bengio, 2014) (see Appendix A.1). Therefore, we can equivalently obtain the score function *at each noise level* by solving the corresponding least squares estimation:

$$\mathbb{E}_{\mathbf{x} \sim p_d} \mathbb{E}_{\mathbf{z} \sim \mathcal{N}(\mathbf{0}, \sigma_t^2 \mathbf{I})} \|r_{\boldsymbol{\theta}}(\hat{\mathbf{x}}; \sigma_t) - \mathbf{x}\|_2^2, \quad \text{where} \quad \hat{\mathbf{x}} = \mathbf{x} + \mathbf{z}. \tag{3}$$

The overall training loss is a weighted combination of Eq. (3) across all noise levels, with the weights reflecting our emphasis on visual quality or density estimation (Song et al., 2021b). Unless otherwise specified, we follow the configuration of EDMs (Karras et al., 2022). In this case, $\mathbf{f}(\mathbf{x}, t) = \mathbf{0}$, $g(t) = \sqrt{2t}$, $\sigma_t = t$, the perturbation kernel $p_t(\hat{\mathbf{x}}|\mathbf{x}) = \mathcal{N}(\hat{\mathbf{x}}; \mathbf{x}, t^2 \mathbf{I})$, and the kernel density estimate $p_t(\hat{\mathbf{x}}) = \int p_d(\mathbf{x}) p_t(\hat{\mathbf{x}}|\mathbf{x}) \mathrm{d}\mathbf{x}$. The optimal estimator for Eq. (3) is given by the conditional expectation $\mathbb{E}(\mathbf{x}|\hat{\mathbf{x}})$, or specifically, $r_{\boldsymbol{\theta}}^{\star}(\hat{\mathbf{x}}; t) = \hat{\mathbf{x}} + t^2 \nabla_{\hat{\mathbf{x}}} \log p_t(\hat{\mathbf{x}})$ as revealed in the literature (Raphan & Simoncelli, 2011; Karras et al., 2022). In practice, we assume that this connection approximately holds after the model training converges, and plug $\nabla_{\mathbf{x}} \log p_t(\mathbf{x}) \approx (r_{\boldsymbol{\theta}}(\mathbf{x}; t) - \mathbf{x})/t^2$ into Eq. (2) to derive the empirical PF-ODE:[1]

$$\mathrm{d}\mathbf{x} = \frac{\mathbf{x} - r_{\boldsymbol{\theta}}(\mathbf{x}; t)}{t} \mathrm{d}t. \tag{4}$$

As for sampling, we first draw $\hat{\mathbf{x}}_{t_N} \sim p_n(\mathbf{x}) = \mathcal{N}(\mathbf{0}, T^2 \mathbf{I})$ and then numerically solve the ODE backwards with $N$ steps to obtain a *sampling trajectory* $\{\hat{\mathbf{x}}_t\}$ with $t \in \{t_0 = 0, t_1, \cdots, t_N = T\}$.[2] The final sample $\hat{\mathbf{x}}_{t_0}$ is considered to approximately follow the data distribution $p_d(\mathbf{x})$.

Besides, we denote another important yet easy to be ignored sequence as $\{r_{\boldsymbol{\theta}}(\hat{\mathbf{x}}_t, t)\}$ or simplified to $\{r_{\boldsymbol{\theta}}(\hat{\mathbf{x}}_t)\}$ if there is no ambiguity, and designate it as *denoising trajectory*. The following proposition reveals that a denoising trajectory is inherently related with the tangent of a sampling trajectory. The visual examples of these two trajectories are provided in the second and third rows of Figure 5.

**Proposition 1.** *The denoising output $r_{\boldsymbol{\theta}}(\mathbf{x}; t)$ reflects the prediction made by a single Euler step from any sample $\mathbf{x}$ at any time toward $t = 0$ with Eq. (4).*

*Proof.* The prediction of such an Euler step equals to $\mathbf{x} + (0 - t)(\mathbf{x} - r_{\boldsymbol{\theta}}(\mathbf{x}; t))/t = r_{\boldsymbol{\theta}}(\mathbf{x}; t)$. $\square$

This property was previously mentioned in (Karras et al., 2022) to advocate the use of Eq. (4) for ODE-based sampling. There, Karras et al. (2022) suspected that this sampling trajectory is approximately linear across most noise levels due to the slow change in denoising output, and verified it in a 1D toy example. In contrast, we provide an in-depth analysis of the high-dimensional trajectory with real data and discover more intriguing structures, especially those related to the denoising trajectory (Sections 3.2 and 4), and reveal a theoretical connection to the classic mean shift (Section 5).

## 3 VISUALIZATION OF HIGH DIMENSIONAL TRAJECTORY

In this section, we present several tools to inspect the trajectory of probability flow ODE in high-dimensional space. We mostly take unconditional generation on CIFAR-10 as an example to demonstrate our observations. The conclusions also hold on other datasets (such as LSUN, ImageNet) and other model settings (such as conditional generation, various network architectures). More results and implementation details are provided in Appendix F.

We adopt $d(\cdot, \cdot)$ to denote the $\ell_2$ distance. Take the sampling trajectory as an example, the distance between a given sample $\hat{\mathbf{x}}_{t_n}$ and the final sample $\hat{\mathbf{x}}_{t_0}$ is denoted as $d(\hat{\mathbf{x}}_{t_n}, \hat{\mathbf{x}}_{t_0})$. The *trajectory deviation* is calculated as the distance between each intermediate sample $\hat{\mathbf{x}}_{t_n}$ and the straight line passing through two endpoints $[\hat{\mathbf{x}}_{t_0} \hat{\mathbf{x}}_{t_N}]$, and denoted as $d(\hat{\mathbf{x}}_{t_n}, [\hat{\mathbf{x}}_{t_0} \hat{\mathbf{x}}_{t_N}])$. The expectation quantities (*e.g.*, distance, magnitude) in every time steps are estimated by averaging 50k generated samples.

---

[1]There seems to be a slight notation ambiguity. Generally, $r_{\boldsymbol{\theta}}(\cdot)$ refers to a converged model in our paper.

[2]The time horizon is divided with the formula $t_n = (t_1^{1/\rho} + \frac{n-1}{N-1}(t_N^{1/\rho} - t_1^{1/\rho}))^\rho$, where $t_1 = 0.002$, $t_N = 80$, $n \in [1, N]$ and $\rho = 7$ (Karras et al., 2022).

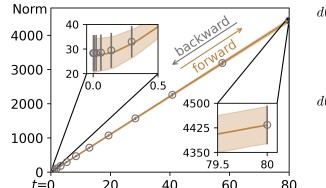 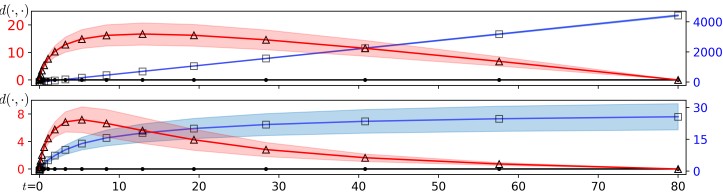

(a) The statistics of magnitude.    (b) The statistics of the sampling (top) and denoising (bottom) trajectories.

Figure 2: (a) The sample magnitude ($\ell_2$ norm) expands in the forward process (brown curve) while shrinking in the backward process (gray circles). (b) Each trajectory deviation (red curve) is calculated as $d(\hat{\mathbf{x}}_{t_n}, [\hat{\mathbf{x}}_{t_0}\hat{\mathbf{x}}_{t_N}])$ or $d(r_{\boldsymbol{\theta}}(\hat{\mathbf{x}}_{t_n}), [r_{\boldsymbol{\theta}}(\hat{\mathbf{x}}_{t_1})r_{\boldsymbol{\theta}}(\hat{\mathbf{x}}_{t_N})])$, respectively, leading to Observation 1. The distance (blue curve) between the final sample and generated intermediate samples is calculated as $d(\hat{\mathbf{x}}_{t_n}, \hat{\mathbf{x}}_{t_0})$ or $d(r_{\boldsymbol{\theta}}(\hat{\mathbf{x}}_{t_n}), r_{\boldsymbol{\theta}}(\hat{\mathbf{x}}_{t_1}))$, respectively, leading to Observation 2.

## 3.1 FORWARD DIFFUSION PROCESS

As discussed in Section 2, the forward process is generally interpreted as a progressive smoothing from data to noise with a series of Gaussian perturbation kernels. We further paraphrase it as the expansion of magnitude and manifold, which means that samples escape from the original *small-magnitude low-rank* manifold and settle into a *large-magnitude high-rank* manifold.

**Proposition 2.** *Given a high-dimensional vector* $\mathbf{x} \in \mathbb{R}^d$ *and an isotropic Gaussian noise* $\mathbf{z} \sim \mathcal{N}(\mathbf{0}; \sigma^2\mathbf{I}_d)$, $\sigma > 0$, *we have* $\mathbb{E} \|\mathbf{z}\|^2 = \sigma^2 d$, *and with high probability,* $\mathbf{z}$ *stays within a "thin shell":* $\|\mathbf{z}\| = \sigma\sqrt{d} \pm O(1)$. *Additionally,* $\mathbb{E} \|\mathbf{x} + \mathbf{z}\|^2 = \|\mathbf{x}\|^2 + \sigma^2 d$, $\lim_{d\to\infty} \mathbb{P}(\|\mathbf{x} + \mathbf{z}\| > \|\mathbf{x}\|) = 1$.

The proofs are provided in Appendix C.2. Proposition 2 implies that in the forward process, the squared magnitude of the noisy sample $\mathbf{x} + \mathbf{z}$ is expected to be larger than that of the original sample $\mathbf{x}$, and their magnitude gap becomes especially huge for the high-dimensional case $d \gg 1$ and severe noise case $\sigma \gg 0$. We can further conclude that as $d \to \infty$, the sample magnitude will expand with probability one and the isotropic Gaussian noise will distribute as a uniform distribution on the sphere (Vershynin, 2018). In practical generative modeling, $d$ is sufficiently large to make this claim approximately correct. The low-rank data manifold is thus lifted to about $d-1$ rank sphere of radius $\sigma\sqrt{d}$, with a thin shell of width $O(1)$. In Figure 2a, we track the magnitude of original data in the forward process and the magnitude of synthetic samples in the backward process. A clear trend is that the sample magnitude expands in the forward diffusion process and shrinks in the backward generative process, and they are well-matched thanks to the marginal preserving property.

## 3.2 BACKWARD GENERATIVE PROCESS

It is challenging to visualize the whole sampling trajectory and the associated denoising trajectory laying in a high-dimensional space. In this paper, we are particularly interested in their geometric properties, and find that each trajectory exhibits a surprisingly simple form. Our observations, which have been confirmed by empirical evidence, are elaborated in the following paragraphs.

**Observation 1.** *The sampling trajectory is almost straight while the denoising trajectory is bent.*

We propose to employ *trajectory deviation* to assess the linearity of trajectories. From Figure 2b, we can see that the deviation of sampling trajectory and denoising trajectory (red curves) gradually increases from $t = 80$ to around $t = 10$ or $t = 5$, respectively, and then quickly decreases until reaching their final samples. This implies that each initial sample may be affected by all possible modes with a large influence at first, and become intensely attracted by its unique mode after a turning point. This phenomenon also supports the strategy of placing time intervals densely near the minimum timestamp yet sparsely near the maximum one (Song et al., 2021a; Karras et al., 2022; Song et al., 2023). However, based on the ratio of maximum deviation (*e.g.*, $\max \mathbb{E}[d(\hat{\mathbf{x}}_{t_n}, [\hat{\mathbf{x}}_{t_0}\hat{\mathbf{x}}_{t_N}])]$) to the endpoint distance (*e.g.*, $\mathbb{E}[d(\hat{\mathbf{x}}_{t_0}, \hat{\mathbf{x}}_{t_N})]$) in Figure 2b, the deviation of sampling trajectory is incredibly small (about $16/4428 \approx 0.0036$), while the deviation of denoising trajectory is relatively significant (about $7/26 \approx 0.27$), which indicates that the former is much straighter than the latter.

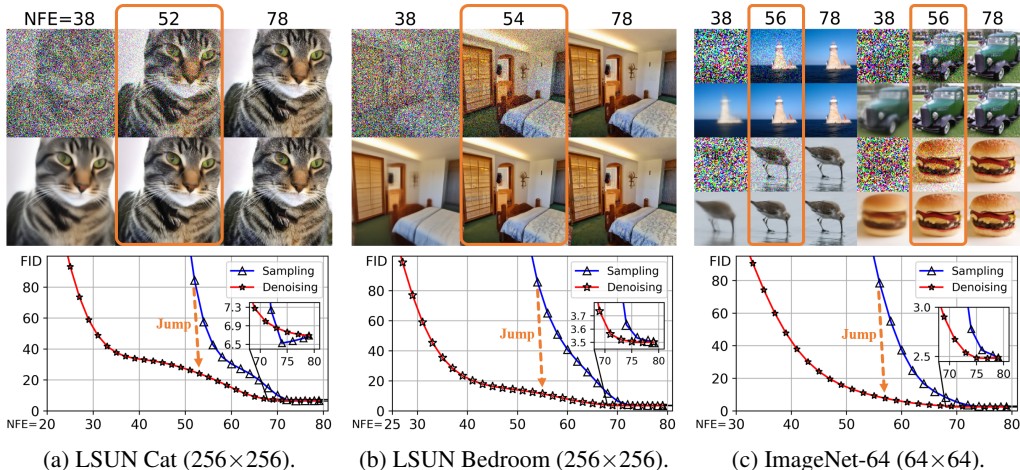

(a) LSUN Cat (256×256).  (b) LSUN Bedroom (256×256).  (c) ImageNet-64 (64×64).

Figure 3: The comparison of visual quality (top is sampling trajectory, bottom is denoising trajectory) and Fréchet Inception Distance (FID (Heusel et al., 2017), lower is better) *w.r.t.* the number of score function evaluations (NFEs). More results are provided in Appendix F.5. The denoising trajectory converges much faster than the sampling trajectory in terms of FID and visual quality.

Another evidence for the quasi-linearity of sampling trajectory is from the aspect of *angle deviation*, which is calculated by the cosine similarity between the backward ODE direction and the direction pointing to the final sample $(\hat{\mathbf{x}}_{t_0} - \hat{\mathbf{x}}_{t_n})$ at the intermediate time $t_n$. We find that $\cos(-\frac{\mathrm{d}\mathbf{x}_t}{\mathrm{d}t}\big|_{t_n}, (\hat{\mathbf{x}}_{t_0} - \hat{\mathbf{x}}_{t_n}))$ always stays in a narrow range from 0.98 to 1.00 (Appendix F.2), which indicates the angle-based trajectory deviation is extremely small and all backward ODE directions almost exactly point to the final sample. Therefore, each initial sample converges monotonically and rapidly by moving along the sampling trajectory, similar to the behavior of gradient descent algorithm in a well-behaved convex function. This claim is confirmed by blue curves in Figure 2b, and summarized as follows

**Observation 2.** *The generated samples on the sampling trajectory and the denoising trajectory both move monotonically from the initial points toward their final points in expectation.*

Observations 1 and 2 enable us to safely adopt large Euler steps or higher-order ODE solvers without incurring much truncation error (Song et al., 2021c; Liu et al., 2022; Karras et al., 2022; Lu et al., 2022). Additionally, we provide the visual quality and FID comparison between the sampling trajectory and the denoising trajectory in Figure 3, and we have the following observation

**Observation 3.** *The denoising trajectory converges faster than the sampling trajectory in terms of visual quality, FID, and sample likelihood.*

The theoretical guarantee *w.r.t* sample likelihood is provided in Section 5 (Theorem 1). This observation inspires us to develop a new sampler named as ODE-Jump that directly jumps from *any* sample at *any* time in the original sampling trajectory simulated by *any* ODE solver to the associated denoising trajectory, and returns the denoising output as the final synthetic image. Specifically, we change the sampling sequence from $\hat{\mathbf{x}}_{t_N} \to \hat{\mathbf{x}}_{t_{N-1}} \to \cdots \to \hat{\mathbf{x}}_{t_n} \to \cdots \to \hat{\mathbf{x}}_{t_1} \to \hat{\mathbf{x}}_{t_0}$ to $\hat{\mathbf{x}}_{t_N} \to \hat{\mathbf{x}}_{t_{N-1}} \to \cdots \to \hat{\mathbf{x}}_{t_n} \to r_{\boldsymbol{\theta}}(\hat{\mathbf{x}}_{t_n})$, and the total NFE reduces from $N$ to $N - n + 1$ if first-order samplers are used. This simple algorithm is highly flexible, extremely easy to implement. We only need to monitor the visual quality of synthetic samples in the implicit denoising trajectory and decide when to interrupt the sampling trajectory and make a jump. Take the sampling on LSUN Bedroom illustrated in Figure 3b as an example, we perform a jump from NFE=54 of the sampling trajectory into NFE=55 of the denoising trajectory and stop the subsequent process. In this step, we achieve a significant FID improvement (from 85.8 to 11.4) and obtain a visually comparable sample with the final one in the original sampling trajectory (NFE=79) at a much less NFE.

All above observations make the picture of ODE-based sampling, as depicted in Figure 1. Geometrically, the initial noise distribution starts from a big sphere and then anisotropically squashes its "radius" and twists the sample range into the exact data manifold. Meanwhile, the distribution of denoising outputs initially approximates a *Dirac delta function* centering in the dataset mean, and

Table 1: Each second-order ODE-based sampler listed below corresponds to a specific finite difference of the denoising trajectory. $\gamma$ denotes a correction coefficient of forward differences. DDIM is a first-order sampler listed for comparison. GENIE trains a neural network to approximate high-order derivatives. $r_{\boldsymbol{\theta}}(\hat{\mathbf{x}}_{t_{n+2}})$ in S-PNDM and DEIS denotes a previous denoising output. $s_n = \sqrt{t_n t_{n+1}}$ in DPM-Solver-2. $\hat{\mathbf{x}}'_{t_n}$ in EDMs denotes the output of an intermediate Euler step.

| ODE solver-based samplers | $\mathrm{d}r_{\boldsymbol{\theta}}(\hat{\mathbf{x}}_{t_{n+1}})/\mathrm{d}t$ | $\gamma$ |
|---|---|---|
| DDIM (Song et al., 2021a) | None | None |
| GENIE (Dockhorn et al., 2022) | Neural Networks | None |
| S-PNDM (Liu et al., 2022) | $\gamma \left(r_{\boldsymbol{\theta}}(\hat{\mathbf{x}}_{t_{n+1}}) - r_{\boldsymbol{\theta}}(\hat{\mathbf{x}}_{t_{n+2}})\right)/(t_n - t_{n+1})$ | 1 |
| DEIS ($\rho$AB1) (Zhang & Chen, 2023) | $\gamma \left(r_{\boldsymbol{\theta}}(\hat{\mathbf{x}}_{t_{n+1}}) - r_{\boldsymbol{\theta}}(\hat{\mathbf{x}}_{t_{n+2}})\right)/(t_{n+1} - t_{n+2})$ | 1 |
| DPM-Solver-2 (Lu et al., 2022) | $\gamma \left(r_{\boldsymbol{\theta}}(\hat{\mathbf{x}}_{s_n}) - r_{\boldsymbol{\theta}}(\hat{\mathbf{x}}_{t_{n+1}})\right)/\left((t_n - t_{n+1})/2\right)$ | $t_{n+1}/s_n$ |
| EDMs (Heun) (Karras et al., 2022) | $\gamma \left(r_{\boldsymbol{\theta}}(\hat{\mathbf{x}}'_{t_n}) - r_{\boldsymbol{\theta}}(\hat{\mathbf{x}}_{t_{n+1}})\right)/(t_n - t_{n+1})$ | $t_{n+1}/t_n$ |

then anisotropically expands its range until exactly matching the data manifold. These two processes are governed by the simple and smooth sampling trajectory and its associated denoising trajectory.

## 4 FINITE DIFFERENCES OF DENOISING TRAJECTORY

To accelerate the sampling speed of diffusion models, various numerical solver-based samplers have been developed in the past several years (Song et al., 2021a;c; Karras et al., 2022; Lu et al., 2022; Zhang & Chen, 2023). In particular, second-order ODE-based samplers are relatively promising in the practical use since they strike a good balance between fast sampling and decent visual quality Rombach et al. (2022); Balaji et al. (2022). More discussion is provided in Appendix D.

In this section, we point out that intriguingly, these prevalent techniques implicitly employ the tangent of denoising trajectory to reduce the truncation error along the sampling trajectory. The probability flow ordinary differential equation of the denoising trajectory is presented as follows

**Proposition 3.** *The ordinary differential equation of the denoising trajectory (denoising-ODE) is*

$$\frac{\mathrm{d}r_{\boldsymbol{\theta}}(\mathbf{x};t)}{\mathrm{d}t} = -t\frac{\mathrm{d}^2\mathbf{x}}{\mathrm{d}t^2}. \tag{5}$$

*Proof.* Since $r_{\boldsymbol{\theta}}(\mathbf{x};t) = \mathbf{x} - t\frac{\mathrm{d}\mathbf{x}}{\mathrm{d}t}$ from Eq. (4), we have $\frac{\mathrm{d}r_{\boldsymbol{\theta}}(\mathbf{x};t)}{\mathrm{d}t} = \frac{\mathrm{d}\mathbf{x}}{\mathrm{d}t} - \left(\frac{\mathrm{d}\mathbf{x}}{\mathrm{d}t} + t\frac{\mathrm{d}^2\mathbf{x}}{\mathrm{d}t^2}\right) = -t\frac{\mathrm{d}^2\mathbf{x}}{\mathrm{d}t^2}$. $\square$

This equation reveals that the denoising trajectory encapsulates the curvature or concavity information of the associated sampling trajectory. Given a sample $\hat{\mathbf{x}}_{t_{n+1}}$, the second-order Taylor polynomial approximation of the sampling trajectory with Eq. (4) is

$$\hat{\mathbf{x}}_{t_n} = \hat{\mathbf{x}}_{t_{n+1}} + \frac{t_n - t_{n+1}}{t_{n+1}}\left(\hat{\mathbf{x}}_{t_{n+1}} - r_{\boldsymbol{\theta}}(\hat{\mathbf{x}}_{t_{n+1}})\right) - \frac{1}{2}\frac{(t_n - t_{n+1})^2}{t_{n+1}}\frac{\mathrm{d}r_{\boldsymbol{\theta}}(\hat{\mathbf{x}}_{t_{n+1}})}{\mathrm{d}t}, \tag{6}$$

where various finite differences of $\mathrm{d}r_{\boldsymbol{\theta}}(\hat{\mathbf{x}}_{t_{n+1}})/\mathrm{d}t$ essentially correspond to a series of second-order samplers, as shown in Table 1. The detailed derivations are provided in Appendix B.

## 5 THEORETICAL CONNECTION TO MEAN SHIFT

Given a parametric diffusion model with the denoising output $r_{\boldsymbol{\theta}}(\cdot)$, the sampling trajectory is simulated by numerically solving Eq. (4), and meanwhile, an implicitly coupled denoising trajectory is formed as a by-product. We next derive the formula of *optimal denoising output* to analyze the asymptotic behavior of diffusion models as they approach the optima.

**Proposition 4.** *The optimal denoising output of Eq. (3) is a convex combination of the original data, where each weight is calculated based on the time-scaled and normalized $\ell_2$ distance between $\hat{\mathbf{x}}$ and*

$\mathbf{x}_i$ *belonging to the dataset* $\mathcal{D}$:

$$r_{\boldsymbol{\theta}}^{\star}(\hat{\mathbf{x}}; \sigma_t) = \sum_i u_i \mathbf{x}_i = \sum_i \frac{\exp\left(-\|\hat{\mathbf{x}} - \mathbf{x}_i\|^2/2\sigma_t^2\right)}{\sum_j \exp\left(-\|\hat{\mathbf{x}} - \mathbf{x}_j\|^2/2\sigma_t^2\right)} \mathbf{x}_i, \quad \sum_i u_i = 1. \tag{7}$$

The proof is provided in Appendix C.3. This equation appears to be highly similar to the well-known non-parametric mean shift (Fukunaga & Hostetler, 1975; Cheng, 1995; Comaniciu & Meer, 2002; Yamasaki & Tanaka, 2020), and we provide a brief overview of it as follows.

Mean shift with a Gaussian kernel and bandwidth $h$ iteratively adds a vector $\mathbf{m}(\mathbf{x}) - \mathbf{x}$, which points toward the maximum increase in the kernel density estimate $p_h(\mathbf{x}) = \frac{1}{N}\sum_i \mathcal{N}(\mathbf{x}; \mathbf{x}_i, h^2\mathbf{I})$, to the current point $\mathbf{x}$, *i.e.*, $\mathbf{x} \leftarrow [\mathbf{m}(\mathbf{x}) - \mathbf{x}] + \mathbf{x}$. The *mean vector* is

$$\mathbf{m}(\mathbf{x}, h) = \sum_i v_i \mathbf{x}_i = \sum_i \frac{\exp\left(-\|\mathbf{x} - \mathbf{x}_i\|^2/2h^2\right)}{\sum_j \exp\left(-\|\mathbf{x} - \mathbf{x}_j\|^2/2h^2\right)} \mathbf{x}_i, \quad \mathbf{x}_i \in \mathcal{D}, \quad \sum_i v_i = 1. \tag{8}$$

From the interpretation of expectation-maximization (EM) algorithm, mean shift converges from almost any initial point with a generally linear convergence rate (Carreira-Perpinan, 2007). As a mode-seeking algorithm, it has shown particularly successful in clustering (Cheng, 1995; Carreira-Perpinán, 2015), image segmentation (Comaniciu & Meer, 2002) and video tracking (Comaniciu et al., 2003). In fact, the ODE-based sampling of diffusion models is closely connected with *annealed mean shift*, or *multi-bandwidth mean shift* (Shen et al., 2005). Annealed mean shift, which was developed as a metaheuristic algorithm for global model seeking, initializes a sufficiently large bandwidth and monotonically decreases it in iterations (Shen et al., 2005). By treating the optimal denoising output as the mean vector in annealed mean shift, we have the following proposition

**Proposition 5.** *Given an optimal probability flow ODE* $\mathrm{d}\mathbf{x} = \frac{\mathbf{x} - r_{\boldsymbol{\theta}}^{\star}(\mathbf{x};t)}{t}\mathrm{d}t$, *one Euler step equals to a convex combination of the annealed mean shift and the current position.*

*Proof.* Given a current sample $\hat{\mathbf{x}}_{t_{n+1}}$, $n \in [0, N-1]$, the prediction of a single Euler step equals to

$$\hat{\mathbf{x}}_{t_n}^{\star} = \hat{\mathbf{x}}_{t_{n+1}} + \frac{t_n - t_{n+1}}{t_{n+1}}\left(\hat{\mathbf{x}}_{t_{n+1}} - r_{\boldsymbol{\theta}}^{\star}(\hat{\mathbf{x}}_{t_{n+1}}; t_{n+1})\right) = \frac{t_n}{t_{n+1}}\hat{\mathbf{x}}_{t_{n+1}} + \frac{t_{n+1} - t_n}{t_{n+1}}\mathbf{m}(\hat{\mathbf{x}}_{t_{n+1}}; t_{n+1}),$$

$$(9)$$

where $\hat{\mathbf{x}}_{t_n}^{\star}$ denotes the generated sample from the optimal PF-ODE, and we treat the discrete time $t_{n+1}$ in $r_{\boldsymbol{\theta}}^{\star}(\hat{\mathbf{x}}_{t_{n+1}}; t_{n+1})$ as the annealing-like bandwidth of Gaussian kernel in Eq. (8). $\square$

Similarly, for the empirical PF-ODE in Eq. (4), each Euler step equals to a convex combination of the denoising output $r_{\boldsymbol{\theta}}(\cdot)$ and the current position. Since the optimal denoising output, or annealed mean shift, starts with a spurious mode (dataset mean) and converges toward a true mode over time, a reasonable choice is to gradually increase its weight in the sampling. In this sense, various time-schedule functions (such as uniform, quadratic, polynomial (Song & Ermon, 2019; Song et al., 2021a; Karras et al., 2022)) essentially boil down to different weighting functions. This interpretation inspires us to directly search proper weights rather than noise schedules with a parametric neural network for better visual quality (Kingma et al., 2021).

Proposition 5 also implies that once a diffusion model has converged to the optimum, all ODE trajectories will be uniquely determined and governed by a bandwidth-varying mean shift. In this case, the forward (encoding) process and backward (decoding) process only depend on the data distribution and the given noise distribution, regardless of model architectures or perturbation kernels. Such a property was previously referred to as *uniquely identifiable encoding* and empirically verified in (Song et al., 2021c), while we theoretically characterize the optimum with annealed mean shift, and thus reveal the asymptotic behavior of diffusion models.

Furthermore, we prove that under a mild condition, the sample likelihood keeps increasing unless $\hat{\mathbf{x}}_{t_n} = r_{\boldsymbol{\theta}}(\hat{\mathbf{x}}_{t_n})$, whether the sample advances along the sampling trajectory or jumps into the denoising trajectory. This offers a theoretical guarantee about our observed geometric structures.

**Theorem 1.** *Suppose that* $\|r_{\boldsymbol{\theta}}^{\star}(\hat{\mathbf{x}}_{t_n}) - r_{\boldsymbol{\theta}}(\hat{\mathbf{x}}_{t_n})\| \leq \|r_{\boldsymbol{\theta}}^{\star}(\hat{\mathbf{x}}_{t_n}) - \hat{\mathbf{x}}_{t_n}\|$ *for a given sample* $\hat{\mathbf{x}}_{t_n}$. *In the ODE-based sampling of diffusion models, the sample likelihood exhibits non-decreasing behavior, i.e.,* $p_h(r_{\boldsymbol{\theta}}(\hat{\mathbf{x}}_{t_n})) \geq p_h(\hat{\mathbf{x}}_{t_n})$ *and* $p_h(\hat{\mathbf{x}}_{t_{n-1}}) \geq p_h(\hat{\mathbf{x}}_{t_n})$ *in terms of the kernel density estimate* $p_h(\mathbf{x}) = \frac{1}{N}\sum_i \mathcal{N}(\mathbf{x}; \mathbf{x}_i, h^2\mathbf{I})$ *with any positive bandwidth* $h$.

The proof is provided in Appendix C.1 and a visual illustration is provided in Figure 4 (top). The assumption requires that our learned denoising output $r_{\boldsymbol{\theta}}(\hat{\mathbf{x}}_{t_n})$ falls within a sphere centered at the optimal denoising output $r_{\boldsymbol{\theta}}^{\star}(\hat{\mathbf{x}}_{t_n})$ with a radius of $\|r_{\boldsymbol{\theta}}^{\star}(\hat{\mathbf{x}}_{t_n}) - \hat{\mathbf{x}}_{t_n}\|$. This radius controls the maximum deviation of the learned denoising output and shrinks during the sampling process. In practice, the assumption is relatively easy to satisfy for a well-trained diffusion model, as shown in Figure 4 (bottom).

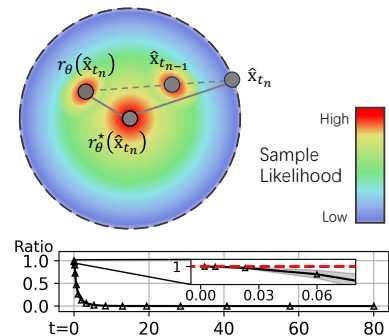

Therefore, each sampling trajectory monotonically converges ($p_h(\hat{\mathbf{x}}_{t_{n-1}}) \geq p_h(\hat{\mathbf{x}}_{t_n})$), and its coupled denoising trajectory converges even faster ($p_h(r_{\boldsymbol{\theta}}(\hat{\mathbf{x}}_{t_n})) \geq p_h(\hat{\mathbf{x}}_{t_n})$) in terms of the sample likelihood. Given an empirical data distribution, Theorem 1 applies to any marginal distributions of our forward SDE $\{p_t(\mathbf{x})\}_{t=0}^{T}$, which are actually a spectrum of kernel density estimates with the positive bandwidth $t$. Besides, with the infinitesimal step size, Theorem 1 is further generalized into a continuous-time version.

Figure 4: The likelihoods of $r_{\boldsymbol{\theta}}(\hat{\mathbf{x}}_{t_n})$ and $\hat{\mathbf{x}}_{t_{n-1}}$ are larger than that of $\hat{\mathbf{x}}_{t_n}$. The ratio of $\|r_{\boldsymbol{\theta}}^{\star}(\hat{\mathbf{x}}_{t_n}) - r_{\boldsymbol{\theta}}(\hat{\mathbf{x}}_{t_n})\|$ to $\|r_{\boldsymbol{\theta}}^{\star}(\hat{\mathbf{x}}_{t_n}) - \hat{\mathbf{x}}_{t_n}\|$ is consistently lower than one in the sampling trajectory.

We can also obtain the well-known monotone convergence property of mean shift, as presented in (Comaniciu & Meer, 2002; Yamasaki & Tanaka, 2020), from Theorem 1 when diffusion models are trained to achieve the optima.

**Corollary 1.** *We have* $p_h(\mathbf{m}(\hat{\mathbf{x}}_{t_n})) \geq p_h(\hat{\mathbf{x}}_{t_n})$, *when* $r_{\boldsymbol{\theta}}(\hat{\mathbf{x}}_{t_n}) = r_{\boldsymbol{\theta}}^{\star}(\hat{\mathbf{x}}_{t_n}) = \mathbf{m}(\hat{\mathbf{x}}_{t_n})$.

## 6 Diagnosis of Score Deviation

We simulate four new trajectories based on the optimal denoising output $r_{\boldsymbol{\theta}}^{\star}(\cdot)$ to monitor the score deviation from the optimum. The first one is *optimal sampling trajectory* $\{\hat{\mathbf{x}}_t^{\star}\}$, where we generate samples as the sampling trajectory $\{\hat{\mathbf{x}}_t\}$ by simulating Eq. (4) but adopt $r_{\boldsymbol{\theta}}^{\star}(\cdot)$ rather than $r_{\boldsymbol{\theta}}(\cdot)$ for score estimation. The other three trajectories are simulated by tracking the (optimal) denoising output of each sample in $\{\hat{\mathbf{x}}_t^{\star}\}$ or $\{\hat{\mathbf{x}}_t\}$, and designated as $\{r_{\boldsymbol{\theta}}(\hat{\mathbf{x}}_t^{\star})\}$, $\{r_{\boldsymbol{\theta}}^{\star}(\hat{\mathbf{x}}_t^{\star})\}$, $\{r_{\boldsymbol{\theta}}^{\star}(\hat{\mathbf{x}}_t)\}$. According to Eq. (9) and $t_0 = 0$, we have $\hat{\mathbf{x}}_{t_0}^{\star} = r_{\boldsymbol{\theta}}^{\star}(\hat{\mathbf{x}}_{t_1}^{\star})$, and similarly, $\hat{\mathbf{x}}_{t_0} = r_{\boldsymbol{\theta}}(\hat{\mathbf{x}}_{t_1})$. As $t \to 0$, $r_{\boldsymbol{\theta}}^{\star}(\hat{\mathbf{x}}_t^{\star})$ and $r_{\boldsymbol{\theta}}^{\star}(\hat{\mathbf{x}}_t)$ serve as the approximate nearest neighbors of $\hat{\mathbf{x}}_t^{\star}$ and $\hat{\mathbf{x}}_t$ to the real data, respectively.

We calculate the deviation of denoising output to quantify the score deviation across all time steps using the $\ell_2$ distance, though they should differ by a factor $t^2$, and have the following observation:

**Observation 4.** *The learned score is well-matched to the optimal score in the large-noise region, otherwise they may diverge or almost coincide depending on different regions.*

In fact, our learned score has to moderately diverge from the optimum to guarantee the generative ability. Otherwise, the ODE-based sampling reduces to an approximate (single-step) annealed mean shift for global mode-seeking (see Section 5), and simply replays the dataset. As shown in Figure 5, the nearest sample of $\hat{\mathbf{x}}_{t_0}^{\star}$ to the real data is almost the same as itself, which indicates the optimal sampling trajectory has a very limited ability to synthesize novel samples. Empirically, score deviation in a small region is sufficient to bring forth a decent generative ability.

From the comparison of $\{r_{\boldsymbol{\theta}}(\hat{\mathbf{x}}_t^{\star})\}$, $\{r_{\boldsymbol{\theta}}^{\star}(\hat{\mathbf{x}}_t^{\star})\}$ sequences in Figures 5 and 6, we can clearly see that *along the optimal sampling trajectory*, the deviation between the learned denoising output $r_{\boldsymbol{\theta}}(\cdot)$ and its optimal counterpart $r_{\boldsymbol{\theta}}^{\star}(\cdot)$ behaves differently in three successive regions: the deviation starts off as almost negligible (about $10 < t \leq 80$), gradually increases (about $3 < t \leq 10$), and then drops down to a low level once again (about $0 \leq t \leq 3$). This phenomenon was also validated by a recent work (Xu et al., 2023) with a different perspective. We further observe that *along the sampling trajectory*, this phenomenon disappears and the score deviation keeps increasing (see $\{r_{\boldsymbol{\theta}}(\hat{\mathbf{x}}_t)\}$, $\{r_{\boldsymbol{\theta}}^{\star}(\hat{\mathbf{x}}_t)\}$ sequences in Figures 5 and 6). Additionally, samples in the latter half of $\{r_{\boldsymbol{\theta}}^{\star}(\hat{\mathbf{x}}_t)\}$ appear almost the same as the nearest sample of $\hat{\mathbf{x}}_{t_0}$ to the real data, as shown in Figure 5. This indicates that our score-based model strives to explore novel regions, and synthetic samples in the sampling trajectory are quickly attracted to a real-data mode but do not fall into it.

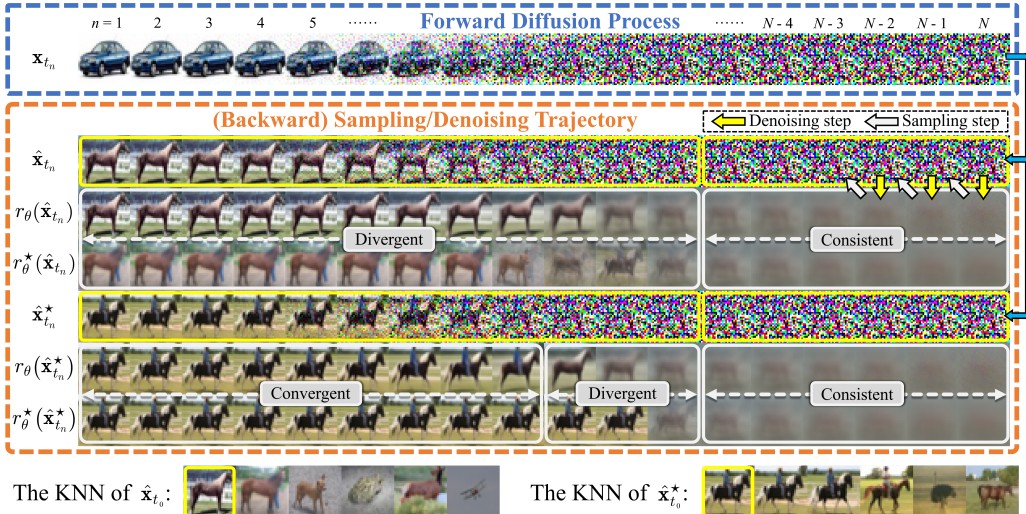

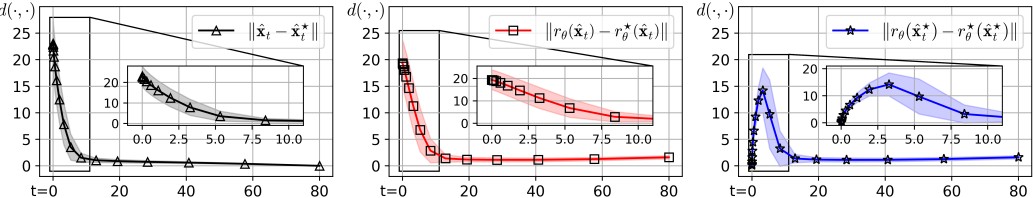

Figure 5: *Top*: We visualize a forward diffusion process of a randomly-selected image to obtain its encoding $\hat{\mathbf{x}}_{t_N}$ (first row) and simulate multiple trajectories starting from this encoding (other rows). *Bottom*: The k-nearest neighbors (k=5) of $\hat{\mathbf{x}}_{t_0}$ and $\hat{\mathbf{x}}_{t_0}^\star$ to real samples in the dataset.

Figure 6: The deviation (measured by $\ell_2$ distance) of outputs from their corresponding optima.

## 7 DISCUSSION

Although all discussions above are provided in the context of VE-SDEs, the similar conclusions also exist for other types of diffusion models (*e.g.*, VP-SDEs). In fact, a family of diffusion models with the same signal-to-noise ratio are closely connected, and we can transform other model types into the VE counterparts with change-of-variables formula (see Appendix A.2). Therefore, we merely focus on the mathematical properties and geometric behaviors of VE-SDEs to simplify our discussions.

## 8 CONCLUSION

In this paper, we present a geometric perspective on (variance-exploding) diffusion models, aiming for a fundamental grasp of their sampling dynamics in an intuitive way. We find that intriguingly, the data distribution and the noise distribution are smoothly bridged by a quasi-linear sampling trajectory and another implicit denoising trajectory that allows faster convergence. These two trajectories are deeply coupled, since each second-order ODE-based sampler along the sampling trajectory corresponds to a specific finite difference of the denoising trajectory. We further characterize the asymptotic behavior of diffusion models by formulating a theoretical relationship between the optimal ODE-based sampling and the anneal mean shift. We hope that our theoretical insights and empirical observations help to better harness the power of score/diffusion-based generative models and facilitate more rapid development in effective training and fast sampling techniques.

**Future work.** The intensively used empirical ODE and its optimal version both behave as a typical non-autonomous non-linear system (Khalil, 2002), which offers a potential approach to discover and analyze more properties (*e.g.*, stability) of the diffusion sampling with tools from control theory.

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

APPENDIX

# A USEFUL RESULTS IN DIFFUSION MODELS

## A.1 AN EMPIRICAL BAYES PERSPECTIVE ON THE CONNECTION BETWEEN DSM AND DAE

In fact, the deep connection between *denoising autoencoder* (DAE) and *denoising score matching* (DSM) already existed within the framework of *empirical Bayes*, which was developed more than half a century ago (Robbins, 1956). We next recap the basic concepts of these terminologies (Vincent et al., 2008; Bengio et al., 2013; Alain & Bengio, 2014; Vincent, 2011; Raphan & Simoncelli, 2011; Efron, 2010; Saremi & Hyvärinen, 2019; Karras et al., 2022) and discuss how they are interconnected, especially in the common setting of diffusion models (Gaussian kernel density estimation). All theoretical results are followed by concise proofs for completion.

Suppose that $\mathbf{x}$ is sampled from an underlying data distribution $p_d(\mathbf{x})$, and the corresponding noisy sample $\hat{\mathbf{x}}$ is generated by a Gaussian perturbation kernel $\hat{\mathbf{x}} \sim p_t(\hat{\mathbf{x}}|\mathbf{x}) = \mathcal{N}(\hat{\mathbf{x}}; \mathbf{x}, \sigma_t^2 \mathbf{I})$ with an isotropic variance $\sigma_t^2$. We can equivalently denote the noisy sample as $\hat{\mathbf{x}} = \mathbf{x} + \mathbf{z}$ with $\mathbf{z} \sim \mathcal{N}(\mathbf{0}, \sigma_t^2 \mathbf{I})$ based on the re-parameterization trick. In diffusion models, we actually have a spectrum of such Gaussian kernels but we just take one of them as an example to simplify the following discussion.

**Lemma 1.** *The optimal estimator $r_{\boldsymbol{\theta}}^{\star}(\hat{\mathbf{x}}; \sigma_t)$, also known as Bayesian least squares estimator, of the denoising autoencoder (DAE) objective is the conditional expectation $\mathbb{E}(\mathbf{x}|\hat{\mathbf{x}})$*

$$
\begin{aligned}
\mathcal{L}_{\mathrm{DAE}} &= \mathbb{E}_{\mathbf{x} \sim p_d(\mathbf{x})} \mathbb{E}_{\mathbf{z} \sim \mathcal{N}(\mathbf{0}, \sigma_t^2 \mathbf{I})} \| r_{\boldsymbol{\theta}}(\hat{\mathbf{x}}; \sigma_t) - \mathbf{x} \|_2^2 \\
&= \int p_d(\mathbf{x}) p_t(\hat{\mathbf{x}}|\mathbf{x}) \| r_{\boldsymbol{\theta}}(\hat{\mathbf{x}}; \sigma_t) - \mathbf{x} \|_2^2 \\
&= \int p_t(\hat{\mathbf{x}}) p_t(\mathbf{x}|\hat{\mathbf{x}}) \| r_{\boldsymbol{\theta}}(\hat{\mathbf{x}}; \sigma_t) - \mathbf{x} \|_2^2.
\end{aligned}
\tag{10}
$$

*Proof.* The solution of this least squares estimation can be easily obtained by setting the derivative of $\mathcal{L}_{\mathrm{DAE}}$ equal to zero. For each noisy sample $\hat{\mathbf{x}}$, we have

$$
\begin{aligned}
\nabla_{r_{\boldsymbol{\theta}}(\hat{\mathbf{x}}; \sigma_t)} \mathcal{L}_{\mathrm{DAE}} &= \mathbf{0} \\
\int p_t(\mathbf{x}|\hat{\mathbf{x}}) \left( r_{\boldsymbol{\theta}}^{\star}(\hat{\mathbf{x}}; \sigma_t) - \mathbf{x} \right) \mathrm{d}\mathbf{x} &= \mathbf{0} \\
\int p_t(\mathbf{x}|\hat{\mathbf{x}}) r_{\boldsymbol{\theta}}^{\star}(\hat{\mathbf{x}}; \sigma_t) \, \mathrm{d}\mathbf{x} &= \int p_t(\mathbf{x}|\hat{\mathbf{x}}) \mathbf{x} \mathrm{d}\mathbf{x} \\
r_{\boldsymbol{\theta}}^{\star}(\hat{\mathbf{x}}; \sigma_t) &= \mathbb{E}(\mathbf{x}|\hat{\mathbf{x}}).
\end{aligned}
\tag{11}
$$

$\square$

**Lemma 2.** *Given a parametric score function $\nabla_{\hat{\mathbf{x}}} \log p_{\theta,t}(\hat{\mathbf{x}})$ of the marginal distribution $p_t(\hat{\mathbf{x}})$, it will converge to the true score function $\nabla_{\hat{\mathbf{x}}} \log p_t(\hat{\mathbf{x}})$ whether it is trained with the denoising score matching (DSM) objective or the explicit score matching (ESM) objective*

$$
\mathcal{L}_{\mathrm{DSM}} = \mathbb{E}_{\mathbf{x} \sim p_d(\mathbf{x})} \mathbb{E}_{\mathbf{z} \sim \mathcal{N}(\mathbf{0}, \sigma_t^2 \mathbf{I})} \| \nabla_{\hat{\mathbf{x}}} \log p_{\theta,t}(\hat{\mathbf{x}}) - \nabla_{\hat{\mathbf{x}}} \log p_t(\hat{\mathbf{x}}|\mathbf{x}) \|_2^2,
\tag{12}
$$

$$
\mathcal{L}_{\mathrm{ESM}} = \mathbb{E}_{\hat{\mathbf{x}} \sim p_t(\hat{\mathbf{x}})} \| \nabla_{\hat{\mathbf{x}}} \log p_{\theta,t}(\hat{\mathbf{x}}) - \nabla_{\hat{\mathbf{x}}} \log p_t(\hat{\mathbf{x}}) \|_2^2.
\tag{13}
$$

*Proof.* Similar to the Lemma 1, the optimal score function trained with the DSM objective is

$$
\begin{aligned}
\nabla_{\hat{\mathbf{x}}} \log p_{\theta,t}^{\star}(\hat{\mathbf{x}}) &= \mathbb{E}_{p_t(\mathbf{x}|\hat{\mathbf{x}})} \nabla_{\hat{\mathbf{x}}} \log p_t(\hat{\mathbf{x}}|\mathbf{x}) \\
&= \int \frac{p_d(\mathbf{x}) p_t(\hat{\mathbf{x}}|\mathbf{x})}{p_t(\hat{\mathbf{x}})} \nabla_{\hat{\mathbf{x}}} \log p_t(\hat{\mathbf{x}}|\mathbf{x}) \mathrm{d}\mathbf{x} \\
&= \int \frac{p_d(\mathbf{x})}{p_t(\hat{\mathbf{x}})} \nabla_{\hat{\mathbf{x}}} p_t(\hat{\mathbf{x}}|\mathbf{x}) \mathrm{d}\mathbf{x} = \frac{\nabla_{\hat{\mathbf{x}}} \int p_d(\mathbf{x}) p_t(\hat{\mathbf{x}}|\mathbf{x}) \mathrm{d}\mathbf{x}}{p_t(\hat{\mathbf{x}})} \\
&= \frac{\nabla_{\hat{\mathbf{x}}} p_t(\hat{\mathbf{x}})}{p_t(\hat{\mathbf{x}})} = \nabla_{\hat{\mathbf{x}}} \log p_t(\hat{\mathbf{x}}).
\end{aligned}
\tag{14}
$$

By setting the derivative of $\mathcal{L}_{\mathrm{ESM}}$ equal to zero, we can also obtain $\nabla_{\hat{\mathbf{x}}} \log p_{\theta,t}^{\star}(\hat{\mathbf{x}}) = \nabla_{\hat{\mathbf{x}}} \log p_t(\hat{\mathbf{x}})$.

$\square$

**Lemma 3.** *The conditional expectation in Lemma 1 can be derived from the score function of the marginal distribution $p_t(\hat{\mathbf{x}})$, i.e., $\mathbb{E}(\mathbf{x}|\hat{\mathbf{x}}) = \hat{\mathbf{x}} + \sigma_t^2 \nabla_{\hat{\mathbf{x}}} \log p_t(\hat{\mathbf{x}})$.*

*Proof.*

$$
\begin{aligned}
p_t(\hat{\mathbf{x}}) &= \int p_d(\mathbf{x}) p_t(\hat{\mathbf{x}}|\mathbf{x}) \mathrm{d}\mathbf{x} \\
\nabla_{\hat{\mathbf{x}}} p_t(\hat{\mathbf{x}}) &= \int \frac{(\mathbf{x} - \hat{\mathbf{x}})}{\sigma_t^2} p_d(\mathbf{x}) p_t(\hat{\mathbf{x}}|\mathbf{x}) \mathrm{d}\mathbf{x} \\
\sigma_t^2 \nabla_{\hat{\mathbf{x}}} p_t(\hat{\mathbf{x}}) &= \int \mathbf{x} p_d(\mathbf{x}) p_t(\hat{\mathbf{x}}|\mathbf{x}) \mathrm{d}\mathbf{x} - \int \hat{\mathbf{x}} p_d(\mathbf{x}) p_t(\hat{\mathbf{x}}|\mathbf{x}) \mathrm{d}\mathbf{x} \\
\sigma_t^2 \nabla_{\hat{\mathbf{x}}} p_t(\hat{\mathbf{x}}) &= \int \mathbf{x} p_d(\mathbf{x}) p_t(\hat{\mathbf{x}}|\mathbf{x}) \mathrm{d}\mathbf{x} - \hat{\mathbf{x}} p_t(\hat{\mathbf{x}}) \\
\sigma_t^2 \frac{\nabla_{\hat{\mathbf{x}}} p_t(\hat{\mathbf{x}})}{p_t(\hat{\mathbf{x}})} &= \int \mathbf{x} p_t(\mathbf{x}|\hat{\mathbf{x}}) \mathrm{d}\mathbf{x} - \hat{\mathbf{x}} \\
\mathbb{E}(\mathbf{x}|\hat{\mathbf{x}}) &= \hat{\mathbf{x}} + \sigma_t^2 \nabla_{\hat{\mathbf{x}}} \log p_t(\hat{\mathbf{x}}).
\end{aligned}
\tag{15}
$$

$\square$

**Remark 1.** *The above intriguing connection implies that we can obtain the posterior expectation $\mathbb{E}(\mathbf{x}|\hat{\mathbf{x}}) = \int \mathbf{x} p_t(\mathbf{x}|\hat{\mathbf{x}}) \mathrm{d}\mathbf{x}$ without explicit accessing to the data density $p_d(\mathbf{x})$ (prior distribution).*

**Remark 2.** *Furthermore, we can estimate the marginal distribution $p_t(\hat{\mathbf{x}})$ or its score function $\nabla_{\hat{\mathbf{x}}} \log p_t(\hat{\mathbf{x}})$ from a set of observed noisy samples to approximate the posterior expectation $\mathbb{E}(\mathbf{x}|\hat{\mathbf{x}})$, or say the optimal denoising output $r_{\boldsymbol{\theta}}^{\star}(\hat{\mathbf{x}}; \sigma_t)$. This exactly shares the philosophy of empirical Bayes that learns a Bayesian prior distribution from the observed data to help statistical inference.*

We provide a brief overview of empirical Bayes as follows.

In contrast to the standard Bayesian methods that specify a *prior* (such as Gaussian, or non-informative distribution) before observing any data, empirical Bayes methods advocate estimating the prior *empirically* from the data itself, whether in a non-parametric or parametric way (Robbins, 1956; Morris, 1983; Efron, 2010; Raphan & Simoncelli, 2011). Using Bayes' theorem, the estimated prior is then converted into the *posterior* by incorporating a *likelihood function* calculated in light of the observed data, as the standard Bayesian methods do.

**Remark 3.** *Learning the marginal distribution in Lemma 3 from the observed data is termed as $f$-modeling in the empirical Bayes literature, in contrast to $g$-modeling that learns the prior distribution from the observed data, see Chapter 21 of a book for more details (Efron & Hastie, 2016).*

**Proposition 6.** *The denoising autoencoder (DAE) objective equals to the denoising score matching (DSM) objective, up to a scaling factor (squared variance), i.e., $\mathcal{L}_{\mathrm{DAE}} = \sigma_t^4 \mathcal{L}_{\mathrm{DSM}}$.*

*Proof.*

$$
\begin{aligned}
\mathcal{L}_{\mathrm{DAE}} &= \mathbb{E}_{\mathbf{x} \sim p_d} \mathbb{E}_{\mathbf{z} \sim \mathcal{N}(\mathbf{0}, \sigma_t^2 \mathbf{I})} \| r_{\boldsymbol{\theta}}(\hat{\mathbf{x}}; \sigma_t) - \mathbf{x} \|_2^2 \\
&= \mathbb{E}_{\mathbf{x} \sim p_d} \mathbb{E}_{\mathbf{z} \sim \mathcal{N}(\mathbf{0}, \sigma_t^2 \mathbf{I})} \| \hat{\mathbf{x}} + \sigma_t^2 \nabla_{\hat{\mathbf{x}}} \log p_{\theta,t}(\hat{\mathbf{x}}) - \mathbf{x} \|_2^2 \\
&= \sigma_t^4 \mathbb{E}_{\mathbf{x} \sim p_d} \mathbb{E}_{\mathbf{z} \sim \mathcal{N}(\mathbf{0}, \sigma_t^2 \mathbf{I})} \| \nabla_{\hat{\mathbf{x}}} \log p_{\theta,t}(\hat{\mathbf{x}}) + \frac{\hat{\mathbf{x}} - \mathbf{x}}{\sigma_t^2} \|_2^2 \\
&= \sigma_t^4 \mathbb{E}_{\mathbf{x} \sim p_d} \mathbb{E}_{\mathbf{z} \sim \mathcal{N}(\mathbf{0}, \sigma_t^2 \mathbf{I})} \| \nabla_{\hat{\mathbf{x}}} \log p_{\theta,t}(\hat{\mathbf{x}}) - \nabla_{\mathbf{x}} \log p_t(\hat{\mathbf{x}}|\mathbf{x}) \|_2^2 \\
&= \sigma_t^4 \mathcal{L}_{\mathrm{DSM}},
\end{aligned}
\tag{16}
$$

where we adopt $\hat{\mathbf{x}} + \sigma_t^2 \nabla_{\hat{\mathbf{x}}} \log p_{\theta,t}(\hat{\mathbf{x}})$ as a parametric empirical Bayes estimator to estimate $\mathbb{E}(\mathbf{x}|\hat{\mathbf{x}})$, which can be compactly written as $r_{\boldsymbol{\theta}}(\hat{\mathbf{x}}; \sigma_t)$. In fact, we have $r_{\boldsymbol{\theta}}(\hat{\mathbf{x}}; \sigma_t) \to r_{\boldsymbol{\theta}}^{\star}(\hat{\mathbf{x}}; \sigma_t)$ as soon as $\nabla_{\hat{\mathbf{x}}} \log p_{\theta,t}(\hat{\mathbf{x}}) \to \nabla_{\hat{\mathbf{x}}} \log p_{\theta,t}^{\star}(\hat{\mathbf{x}}) = \nabla_{\hat{\mathbf{x}}} \log p_t(\hat{\mathbf{x}})$, and vice versa. $\square$

**Remark 4.** *The relationship between denoising autoencoder (DAE), denoising score matching (DSM), and empirical Bayes are summarized as*

$$
\underbrace{r_{\boldsymbol{\theta}}(\hat{\mathbf{x}}; \sigma_t) \to r_{\boldsymbol{\theta}}^{\star}(\hat{\mathbf{x}}; \sigma_t)}_{\mathrm{DAE}} = \overbrace{\mathbb{E}(\mathbf{x}|\hat{\mathbf{x}}) = \hat{\mathbf{x}} + \sigma_t^2 \nabla_{\hat{\mathbf{x}}} \log p_t(\hat{\mathbf{x}})}^{\mathrm{Empirical\ Bayes}} \leftarrow \underbrace{\hat{\mathbf{x}} + \sigma_t^2 \nabla_{\hat{\mathbf{x}}} \log p_{\theta,t}(\hat{\mathbf{x}})}_{\mathrm{DSM}}.
\tag{17}
$$

## A.2 THE CHANGE-OF-VARIABLES FORMULA IN DIFFUSION MODELS

In this section, we show that various diffusion models sharing the same signal-to-noise ratio (SNR) are closely connected with the change-of-variables formula. In particular, all other model types (*e.g.*, variance-preserving (VP) diffusion process used in DDPMs (Ho et al., 2020)) can be transformed into the variance-exploding (VE) counterparts. Therefore, we merely focus on the mathematical properties and geometric behaviors of VE-SDEs in the main text to simplify our discussions.

We first recap the widely used forward SDEs in diffusion models and their basic notations as follows

$$\mathrm{d}\mathbf{x} = \mathbf{f}(t)\mathbf{x}\mathrm{d}t + g(t)\mathrm{d}\mathbf{w}_t, \qquad \mathbf{f}(\cdot) : \mathbb{R} \to \mathbb{R}, \quad g(\cdot) : \mathbb{R} \to \mathbb{R}, \tag{18}$$

where $\mathbf{w}_t$ is the standard Wiener process; $\mathbf{f}(t)$ and $g(t)$ are drift and diffusion coefficients, respectively (Oksendal, 2013; Song et al., 2021c). The perturbation kernels

$$p_{0t}\left(\mathbf{x}_t|\mathbf{x}_0\right) = \mathcal{N}\left(\mathbf{x}_t; s(t)\mathbf{x}_0, s(t)^2\sigma(t)^2\mathbf{I}\right), \tag{19}$$

are derived with the standard techniques in differential equations (Karras et al., 2022) and the coefficients $s(t)$ and $\sigma(t)$ in Eq. (19) are expressed with drift and diffusion coefficients $f(t)$ and $g(t)$:

$$s(t) = \exp\left(\int_0^t f(\xi)\mathrm{d}\xi\right), \quad \text{and} \quad \sigma(t) = \sqrt{\int_0^t \left[\frac{g(\xi)}{s(\xi)}\right]^2 \mathrm{d}\xi}. \tag{20}$$

Or equivalently, we can rewrite drift and diffusion coefficients $f(t)$ and $g(t)$ with $s(t)$ and $\sigma(t)$:

$$f(t) = \frac{\mathrm{d}\log s(t)}{\mathrm{d}t}, \quad \text{and} \quad g(t) = s(t)\sqrt{\frac{\mathrm{d}\left[\sigma^2(t)\right]}{\mathrm{d}t}}. \tag{21}$$

Therefore, the forward SDEs in Eq. (18) can be alternatively expressed in terms of $s(t)$ and $\sigma(t)$:

$$\mathrm{d}\mathbf{x} = \frac{\mathrm{d}\log s(t)}{\mathrm{d}t}\mathbf{x}\,\mathrm{d}t + s(t)\sqrt{\frac{\mathrm{d}\left[\sigma^2(t)\right]}{\mathrm{d}t}}\,\mathrm{d}\mathbf{w}_t. \tag{22}$$

An importance implication from Eq. (19) is that different diffusion models sharing the same $\sigma(t)$ actually have the same signal-to-noise ratio (SNR), since SNR $= s(t)^2/\left[s(t)^2\sigma(t)^2\right] = 1/\left[\sigma(t)^2\right]$.

The standard notions of VP-SDEs (Ho et al., 2020; Song et al., 2021a;c) can be recovered by setting $s(t) = \sqrt{\alpha(t)}$, $\sigma(t) = \sqrt{\frac{1-\alpha(t)}{\alpha(t)}}$ in Eq. (19) and Eq. (22), and $\beta(t) = -\frac{1}{\alpha(t)}\frac{\mathrm{d}\alpha(t)}{\mathrm{d}t}$. We then have

$$\mathbf{x}_t = \sqrt{\alpha(t)}\,\mathbf{x}_0 + \sqrt{1-\alpha(t)}\,\boldsymbol{\epsilon}, \quad \boldsymbol{\epsilon} \sim \mathcal{N}(\mathbf{0},\mathbf{I}),$$

$$\mathrm{d}\mathbf{x} = \frac{1}{2\alpha(t)}\frac{\mathrm{d}\alpha(t)}{\mathrm{d}t}\mathbf{x}\,\mathrm{d}t + \sqrt{-\frac{1}{\alpha(t)}\frac{\mathrm{d}\alpha(t)}{\mathrm{d}t}}\,\mathrm{d}\mathbf{w}_t, \tag{23}$$

$$\mathrm{d}\mathbf{x} = -\frac{1}{2}\beta(t)\mathbf{x}\,\mathrm{d}t + \sqrt{\beta(t)}\mathbf{w}_t.$$

Similarly, the standard notions of VE-SDEs (Song & Ermon, 2019; Song et al., 2021c) can be recovered by setting $s(t) = 1$, and we have

$$\mathbf{x}_t = \mathbf{x}_0 + \sigma(t)\boldsymbol{\epsilon}, \quad \boldsymbol{\epsilon} \sim \mathcal{N}(\mathbf{0},\mathbf{I}),$$

$$\mathrm{d}\mathbf{x} = \sqrt{\frac{\mathrm{d}\left[\sigma^2(t)\right]}{\mathrm{d}t}}\,\mathrm{d}\mathbf{w}_t. \tag{24}$$

The above examples inspire us to eliminate the effect of $s(t)$ if we hope to transform any other types of diffusion models into the VE counterparts, *i.e.*, $f(t) = 0$.

**Proposition 7.** *A diffusion model defined as Eq. (22) can be transformed into its VE counterpart with the change-of-variables formula $\bar{\mathbf{x}} = \mathbf{x}/s(t)$, keeping the SNR unchanged.*

*Proof.* By setting $\bar{\mathbf{x}} = \mathbf{x}/s(t)$ in Eq. (22), we have

$$\frac{\mathrm{d}\left[\bar{\mathbf{x}}s(t)\right]}{\mathrm{d}t} = \frac{\mathrm{d}\log s(t)}{\mathrm{d}t}\left[\bar{\mathbf{x}}s(t)\right] + s(t)\sqrt{\frac{\mathrm{d}\left[\sigma^2(t)\right]}{\mathrm{d}t}}\frac{\mathrm{d}\mathbf{w}_t}{\mathrm{d}t}$$

$$\frac{\mathrm{d}\bar{\mathbf{x}}}{\mathrm{d}t}s(t) + \frac{\mathrm{d}s(t)}{\mathrm{d}t}\bar{\mathbf{x}} = \frac{\mathrm{d}\log s(t)}{\mathrm{d}t}\left[\bar{\mathbf{x}}s(t)\right] + s(t)\sqrt{\frac{\mathrm{d}\left[\sigma^2(t)\right]}{\mathrm{d}t}}\frac{\mathrm{d}\mathbf{w}_t}{\mathrm{d}t}, \tag{25}$$

$$\mathrm{d}\bar{\mathbf{x}} = \sqrt{\frac{\mathrm{d}\left[\sigma^2(t)\right]}{\mathrm{d}t}}\,\mathrm{d}\mathbf{w}_t.$$

$\square$

In the above derivation, the $\sigma(t)$ in the new VE-SDE ($\bar{\mathbf{x}}$-space) is exactly the same as the $\sigma(t)$ used in the original SDE ($\mathbf{x}$-space), and thus the SNR remains unchanged.

# B SECOND-ORDER ODE SAMPLERS AS FINITE DIFFERENCES OF DENOISING TRAJECTORY

In the past several years, various second-order ODE-based samplers have been developed to accelerate the sampling speed of diffusion models (Song et al., 2021a;c; Karras et al., 2022; Lu et al., 2022; Zhang & Chen, 2023). We next demonstrate that each of them can be rewritten as a specific way to perform finite difference of the denoising trajectory. The PF-ODEs of the sampling trajectory Eq. (4) and denoising trajectory Eq. (5) are provided as follows

$$\text{sampling-ODE: } \frac{\mathrm{d}\mathbf{x}}{\mathrm{d}t} = \boldsymbol{\epsilon}(\mathbf{x};t) = \frac{\mathbf{x} - r_{\boldsymbol{\theta}}(\mathbf{x};t)}{t}, \quad \text{denoising-ODE: } \frac{\mathrm{d}r_{\boldsymbol{\theta}}(\mathbf{x};t)}{\mathrm{d}t} = -t\frac{\mathrm{d}^2\mathbf{x}}{\mathrm{d}t^2},$$

where we introduce the notation $\boldsymbol{\epsilon}(\mathbf{x};t)$ to simplify the following derivations. Given a current sample $\hat{\mathbf{x}}_{t_{n+1}}$, the second-order Taylor polynomial approximation of the sampling trajectory is

$$
\begin{aligned}
\hat{\mathbf{x}}_{t_n} &= \hat{\mathbf{x}}_{t_{n+1}} + (t_n - t_{n+1})\frac{\mathrm{d}\mathbf{x}}{\mathrm{d}t}\Big|_{\hat{\mathbf{x}}_{t_{n+1}}} + \frac{1}{2}(t_n - t_{n+1})^2\frac{\mathrm{d}^2\mathbf{x}}{\mathrm{d}t^2}\Big|_{\hat{\mathbf{x}}_{t_{n+1}}} \\
&= \hat{\mathbf{x}}_{t_{n+1}} + \frac{t_n - t_{n+1}}{t_{n+1}}\left(\hat{\mathbf{x}}_{t_{n+1}} - r_{\boldsymbol{\theta}}(\hat{\mathbf{x}}_{t_{n+1}})\right) - \frac{1}{2}\frac{(t_n - t_{n+1})^2}{t_{n+1}}\frac{\mathrm{d}r_{\boldsymbol{\theta}}(\hat{\mathbf{x}}_{t_{n+1}})}{\mathrm{d}t}.
\end{aligned}
\tag{26}
$$

Or, the above equation can be written with the notion of $\boldsymbol{\epsilon}(\mathbf{x};t)$

$$\hat{\mathbf{x}}_{t_n} = \hat{\mathbf{x}}_{t_{n+1}} + (t_n - t_{n+1})\boldsymbol{\epsilon}_{\boldsymbol{\theta}}(\hat{\mathbf{x}}_{t_{n+1}}) + \frac{1}{2}(t_n - t_{n+1})^2\frac{\mathrm{d}\boldsymbol{\epsilon}_{\boldsymbol{\theta}}(\hat{\mathbf{x}}_{t_{n+1}})}{\mathrm{d}t}.\tag{27}$$

All following proofs are conducted in the context of our VE-SDE, i.e., $\mathrm{d}\mathbf{x} = \sqrt{2t}\,\mathrm{d}\mathbf{w}_t$, and the sampling trajectory always starts from $\hat{\mathbf{x}}_{t_N} \sim \mathcal{N}(\mathbf{0}, T^2\mathbf{I})$ and ends at $\hat{\mathbf{x}}_{t_0}$.

## B.1 EDMs (KARRAS ET AL., 2022)

EDMs employ Heun's 2$^{\text{nd}}$ order method, where one Euler step is first applied and followed by a second order correction, which can be written as

$$
\begin{aligned}
\hat{\mathbf{x}}'_{t_n} &= \hat{\mathbf{x}}_{t_{n+1}} + (t_n - t_{n+1})\boldsymbol{\epsilon}_{\boldsymbol{\theta}}(\hat{\mathbf{x}}_{t_{n+1}}), \\
\hat{\mathbf{x}}_{t_n} &= \hat{\mathbf{x}}_{t_{n+1}} + (t_n - t_{n+1})\left(0.5\boldsymbol{\epsilon}_{\boldsymbol{\theta}}(\hat{\mathbf{x}}_{t_{n+1}}) + 0.5\boldsymbol{\epsilon}_{\boldsymbol{\theta}}(\hat{\mathbf{x}}'_{t_n})\right) \\
&= \hat{\mathbf{x}}_{t_{n+1}} + (t_n - t_{n+1})\boldsymbol{\epsilon}_{\boldsymbol{\theta}}(\hat{\mathbf{x}}_{t_{n+1}}) + \frac{1}{2}(t_n - t_{n+1})^2\frac{\boldsymbol{\epsilon}_{\boldsymbol{\theta}}(\hat{\mathbf{x}}'_{t_n}) - \boldsymbol{\epsilon}_{\boldsymbol{\theta}}(\hat{\mathbf{x}}_{t_{n+1}})}{t_n - t_{n+1}}.
\end{aligned}
\tag{28}
$$

By using $r_{\boldsymbol{\theta}}(\mathbf{x};t) = \mathbf{x} - t\boldsymbol{\epsilon}_{\boldsymbol{\theta}}(\mathbf{x};t)$, the sampling iteration above is equivalent to

$$
\begin{aligned}
\hat{\mathbf{x}}_{t_n} &= \hat{\mathbf{x}}_{t_{n+1}} + (t_n - t_{n+1})\frac{\hat{\mathbf{x}}_{t_{n+1}} - r_{\boldsymbol{\theta}}(\hat{\mathbf{x}}_{t_{n+1}})}{t_{n+1}} + \frac{1}{2}(t_n - t_{n+1})^2\frac{\frac{\hat{\mathbf{x}}'_{t_n} - r_{\boldsymbol{\theta}}(\hat{\mathbf{x}}'_{t_n})}{t_n} - \frac{\hat{\mathbf{x}}_{t_{n+1}} - r_{\boldsymbol{\theta}}(\hat{\mathbf{x}}_{t_{n+1}})}{t_{n+1}}}{t_n - t_{n+1}} \\
&= \frac{t_n}{t_{n+1}}\hat{\mathbf{x}}_{t_{n+1}} + \frac{t_{n+1} - t_n}{t_{n+1}}r_{\boldsymbol{\theta}}(\hat{\mathbf{x}}_{t_{n+1}}) - \frac{1}{2}\frac{(t_n - t_{n+1})^2}{t_{n+1}}\frac{t_{n+1}}{t_n}\frac{r_{\boldsymbol{\theta}}(\hat{\mathbf{x}}'_{t_n}) - r_{\boldsymbol{\theta}}(\hat{\mathbf{x}}_{t_{n+1}})}{t_n - t_{n+1}}.
\end{aligned}
\tag{29}
$$

## B.2 DPM-SOLVER (LU ET AL., 2022)

According to (3.3) in DPM-Solver, the exact solution of PF-ODE in the VE-SDE setting is given by

$$\hat{\mathbf{x}}_{t_n} = \hat{\mathbf{x}}_{t_{n+1}} + \int_{t_{n+1}}^{t_n} \boldsymbol{\epsilon}_{\boldsymbol{\theta}}(\hat{\mathbf{x}}_t)\mathrm{d}t.\tag{30}$$

The Taylor expansion of $\boldsymbol{\epsilon}_{\boldsymbol{\theta}}(\hat{\mathbf{x}}_t)$ w.r.t. time $t$ at $t_{n+1}$ is

$$\boldsymbol{\epsilon}_{\boldsymbol{\theta}}(\hat{\mathbf{x}}_t) = \sum_{m=0}^{k-1} \frac{(t - t_{n+1})^m}{m!}\boldsymbol{\epsilon}_{\boldsymbol{\theta}}^{(m)}(\hat{\mathbf{x}}_{t_{n+1}}) + \mathcal{O}\left((t - t_{n+1})^k\right).\tag{31}$$

Then, the DPM-Solver-k sampler can be written as

$$\hat{\mathbf{x}}_{t_n} = \hat{\mathbf{x}}_{t_{n+1}} + \sum_{m=0}^{k-1} \boldsymbol{\epsilon}_{\boldsymbol{\theta}}^{(m)}(\hat{\mathbf{x}}_{t_{n+1}}) \int_{t_{n+1}}^{t_n} \frac{(t - t_{n+1})^m}{m!} \mathrm{d}t + \mathcal{O}\left((t_n - t_{n+1})^k\right)$$

$$= \hat{\mathbf{x}}_{t_{n+1}} + \sum_{m=0}^{k-1} \frac{(t_n - t_{n+1})^{m+1}}{(m+1)!} \boldsymbol{\epsilon}_{\boldsymbol{\theta}}^{(m)}(\hat{\mathbf{x}}_{t_{n+1}}) + \mathcal{O}\left((t_n - t_{n+1})^{k+1}\right). \tag{32}$$

Specifically, when $k = 2$, the DPM-Solver-2 sampler is given by

$$\hat{\mathbf{x}}_{t_n} = \hat{\mathbf{x}}_{t_{n+1}} + (t_n - t_{n+1})\boldsymbol{\epsilon}_{\boldsymbol{\theta}}(\hat{\mathbf{x}}_{t_{n+1}}) + \frac{1}{2}(t_n - t_{n+1})^2 \frac{\mathrm{d}\boldsymbol{\epsilon}_{\boldsymbol{\theta}}(\hat{\mathbf{x}}_{t_{n+1}})}{\mathrm{d}t}, \tag{33}$$

which is the same as Eq. (27). The above second-order term in (Lu et al., 2022) is approximated by

$$\frac{\mathrm{d}\boldsymbol{\epsilon}_{\boldsymbol{\theta}}(\hat{\mathbf{x}}_{t_{n+1}})}{\mathrm{d}t} \approx \frac{\boldsymbol{\epsilon}_{\boldsymbol{\theta}}(\hat{\mathbf{x}}_{s_n}) - \boldsymbol{\epsilon}_{\boldsymbol{\theta}}(\hat{\mathbf{x}}_{t_{n+1}})}{(t_n - t_{n+1})/2}, \tag{34}$$

where $s_n = \sqrt{t_n t_{n+1}}$ and $\hat{\mathbf{x}}_{s_n} = \hat{\mathbf{x}}_{t_{n+1}} + (s_n - t_{n+1})\boldsymbol{\epsilon}_{\boldsymbol{\theta}}(\hat{\mathbf{x}}_{t_{n+1}})$. We have

$$\hat{\mathbf{x}}_{t_n} = \hat{\mathbf{x}}_{t_{n+1}} + (t_n - t_{n+1})\boldsymbol{\epsilon}_{\boldsymbol{\theta}}(\hat{\mathbf{x}}_{t_{n+1}}) + \frac{1}{2}(t_n - t_{n+1})^2 \frac{\boldsymbol{\epsilon}_{\boldsymbol{\theta}}(\hat{\mathbf{x}}_{s_n}) - \boldsymbol{\epsilon}_{\boldsymbol{\theta}}(\hat{\mathbf{x}}_{t_{n+1}})}{(t_n - t_{n+1})/2}$$

$$= \frac{t_n}{t_{n+1}}\hat{\mathbf{x}}_{t_{n+1}} + \frac{t_{n+1} - t_n}{t_{n+1}}r_{\boldsymbol{\theta}}(\hat{\mathbf{x}}_{t_{n+1}}) - \frac{1}{2}\frac{(t_n - t_{n+1})^2}{t_{n+1}}\frac{t_{n+1}}{s_n}\frac{r_{\boldsymbol{\theta}}(\hat{\mathbf{x}}_{s_n}) - r_{\boldsymbol{\theta}}(\hat{\mathbf{x}}_{t_{n+1}})}{(t_n - t_{n+1})/2}. \tag{35}$$

### B.3 PNDM (LIU ET AL., 2022)

Assume that the previous denoising output $r_{\boldsymbol{\theta}}(\hat{\mathbf{x}}_{t_{n+2}})$ is available, then one S-PNDM sampler step can be written as

$$\hat{\mathbf{x}}_{t_n} = \hat{\mathbf{x}}_{t_{n+1}} + (t_n - t_{n+1})\frac{1}{2}\left(3\boldsymbol{\epsilon}_{\boldsymbol{\theta}}(\hat{\mathbf{x}}_{t_{n+1}}) - \boldsymbol{\epsilon}_{\boldsymbol{\theta}}(\hat{\mathbf{x}}_{t_{n+2}})\right)$$

$$= \hat{\mathbf{x}}_{t_{n+1}} + (t_n - t_{n+1})\boldsymbol{\epsilon}_{\boldsymbol{\theta}}(\hat{\mathbf{x}}_{t_{n+1}}) + \frac{1}{2}(t_n - t_{n+1})^2 \frac{\boldsymbol{\epsilon}_{\boldsymbol{\theta}}(\hat{\mathbf{x}}_{t_{n+1}}) - \boldsymbol{\epsilon}_{\boldsymbol{\theta}}(\hat{\mathbf{x}}_{t_{n+2}})}{t_n - t_{n+1}}$$

$$= \frac{t_n}{t_{n+1}}\hat{\mathbf{x}}_{t_{n+1}} + \frac{t_{n+1} - t_n}{t_{n+1}}r_{\boldsymbol{\theta}}(\hat{\mathbf{x}}_{t_{n+1}}) - \frac{1}{2}\frac{(t_n - t_{n+1})^2}{t_{n+1}}\frac{r_{\boldsymbol{\theta}}(\hat{\mathbf{x}}_{t_{n+1}}) - r_{\boldsymbol{\theta}}(\hat{\mathbf{x}}_{t_{n+2}})}{t_n - t_{n+1}}. \tag{36}$$

### B.4 DEIS (ZHANG & CHEN, 2023)

In DEIS paper, the solution of PF-ODE in the VE-SDE setting is given by

$$\hat{\mathbf{x}}_{t_n} = \hat{\mathbf{x}}_{t_{n+1}} + \sum_{j=0}^{r} C_{ij}\boldsymbol{\epsilon}_{\boldsymbol{\theta}}(\hat{\mathbf{x}}_{t_{n+1+j}}),$$

$$C_{ij} = \int_{t_{n+1}}^{t_n} \prod_{k \neq j} \frac{\tau - t_{n+1+k}}{t_{n+1+j} - t_{n+1+k}} \mathrm{d}\tau. \tag{37}$$

$r = 1$ yields the $\rho$AB1 sampler:

$$\hat{\mathbf{x}}_{t_n} = \hat{\mathbf{x}}_{t_{n+1}} + \boldsymbol{\epsilon}_{\boldsymbol{\theta}}(\hat{\mathbf{x}}_{t_{n+1}}) \int_{t_{n+1}}^{t_n} \frac{\tau - t_{n+2}}{t_{n+1} - t_{n+2}} \mathrm{d}\tau + \boldsymbol{\epsilon}_{\boldsymbol{\theta}}(\hat{\mathbf{x}}_{t_{n+2}}) \int_{t_{n+1}}^{t_n} \frac{\tau - t_{n+1}}{t_{n+2} - t_{n+1}} \mathrm{d}\tau$$

$$= \hat{\mathbf{x}}_{t_{n+1}} + \boldsymbol{\epsilon}_{\boldsymbol{\theta}}(\hat{\mathbf{x}}_{t_{n+1}}) \frac{(t_n - t_{n+2})^2 - (t_{n+1} - t_{n+2})^2}{2(t_{n+1} - t_{n+2})} + \boldsymbol{\epsilon}_{\boldsymbol{\theta}}(\hat{\mathbf{x}}_{t_{n+2}}) \frac{(t_n - t_{n+1})^2}{2(t_{n+2} - t_{n+1})}$$

$$= \hat{\mathbf{x}}_{t_{n+1}} + (t_n - t_{n+1})\boldsymbol{\epsilon}_{\boldsymbol{\theta}}(\hat{\mathbf{x}}_{t_{n+1}}) + \frac{1}{2}(t_n - t_{n+1})^2 \frac{\boldsymbol{\epsilon}_{\boldsymbol{\theta}}(\hat{\mathbf{x}}_{t_{n+1}}) - \boldsymbol{\epsilon}_{\boldsymbol{\theta}}(\hat{\mathbf{x}}_{t_{n+2}})}{t_{n+1} - t_{n+2}}$$

$$= \frac{t_n}{t_{n+1}}\hat{\mathbf{x}}_{t_{n+1}} + \frac{t_{n+1} - t_n}{t_{n+1}}r_{\boldsymbol{\theta}}(\hat{\mathbf{x}}_{t_{n+1}}) - \frac{1}{2}\frac{(t_n - t_{n+1})^2}{t_{n+1}}\frac{r_{\boldsymbol{\theta}}(\hat{\mathbf{x}}_{t_{n+1}}) - r_{\boldsymbol{\theta}}(\hat{\mathbf{x}}_{t_{n+2}})}{t_{n+1} - t_{n+2}}. \tag{38}$$

## C PROOFS OF THEOREM AND PROPOSITIONS

### C.1 MONOTONE INCREASE IN SAMPLE LIKELIHOOD (THEOREM 1 IN THE MAIN TEXT)

**Theorem 1.** *Suppose that $\|r_{\boldsymbol{\theta}}^{\star}(\hat{\mathbf{x}}_{t_n}) - r_{\boldsymbol{\theta}}(\hat{\mathbf{x}}_{t_n})\| \leq \|r_{\boldsymbol{\theta}}^{\star}(\hat{\mathbf{x}}_{t_n}) - \hat{\mathbf{x}}_{t_n}\|$ for a given sample $\hat{\mathbf{x}}_{t_n}$. In the ODE-based sampling of diffusion models, the sample likelihood exhibits non-decreasing behavior, i.e., $p_h(r_{\boldsymbol{\theta}}(\hat{\mathbf{x}}_{t_n})) \geq p_h(\hat{\mathbf{x}}_{t_n})$ and $p_h(\hat{\mathbf{x}}_{t_{n-1}}) \geq p_h(\hat{\mathbf{x}}_{t_n})$ in terms of the kernel density estimate $p_h(\mathbf{x}) = \frac{1}{N} \sum_i \mathcal{N}(\mathbf{x}; \mathbf{x}_i, h^2 \mathbf{I})$ with any positive bandwidth $h$.*

*Proof.* We first prove that given a random vector $\mathbf{v}$ falling within a sphere centered at the optimal denoising output $r_{\boldsymbol{\theta}}^{\star}(\hat{\mathbf{x}}_{t_n})$ with a radius of $\|r_{\boldsymbol{\theta}}^{\star}(\hat{\mathbf{x}}_{t_n}) - \hat{\mathbf{x}}_{t_n}\|$, i.e., $\|r_{\boldsymbol{\theta}}^{\star}(\hat{\mathbf{x}}_{t_n}) - \hat{\mathbf{x}}_{t_n}\| \geq \|\mathbf{v}\|$, the sample likelihood is non-decreasing from $\hat{\mathbf{x}}_{t_n}$ to $r_{\boldsymbol{\theta}}^{\star}(\hat{\mathbf{x}}_{t_n}) + \mathbf{v}$, i.e., $p_h(r_{\boldsymbol{\theta}}^{\star}(\hat{\mathbf{x}}_{t_n}) + \mathbf{v}) \geq p_h(\hat{\mathbf{x}}_{t_n})$. Then, we provide two settings for $\mathbf{v}$ to finish the proof.

The increase of sample likelihood from $\hat{\mathbf{x}}_{t_n}$ to $r_{\boldsymbol{\theta}}^{\star}(\hat{\mathbf{x}}_{t_n}) + \mathbf{v}$ in terms of $p_h(\mathbf{x})$ is

$$p_h(r_{\boldsymbol{\theta}}^{\star}(\hat{\mathbf{x}}_{t_n}) + \mathbf{v}) - p_h(\hat{\mathbf{x}}_{t_n}) = \frac{1}{N} \sum_i \left[ \mathcal{N}\left(r_{\boldsymbol{\theta}}^{\star}(\hat{\mathbf{x}}_{t_n}) + \mathbf{v}; \mathbf{x}_i, h^2 \mathbf{I}\right) - \mathcal{N}\left(\hat{\mathbf{x}}_{t_n}; \mathbf{x}_i, h^2 \mathbf{I}\right) \right]$$

$$\overset{(i)}{\geq} \frac{1}{2h^2 N} \sum_i \mathcal{N}\left(\hat{\mathbf{x}}_{t_n}; \mathbf{x}_i, h^2 \mathbf{I}\right) \left[ \|\hat{\mathbf{x}}_{t_n} - \mathbf{x}_i\|^2 - \|r_{\boldsymbol{\theta}}^{\star}(\hat{\mathbf{x}}_{t_n}) + \mathbf{v} - \mathbf{x}_i\|^2 \right]$$

$$= \frac{1}{2h^2 N} \sum_i \mathcal{N}\left(\hat{\mathbf{x}}_{t_n}; \mathbf{x}_i, h^2 \mathbf{I}\right) \left[ \|\hat{\mathbf{x}}_{t_n}\|^2 - 2\hat{\mathbf{x}}_{t_n}^T \mathbf{x}_i - \|r_{\boldsymbol{\theta}}^{\star}(\hat{\mathbf{x}}_{t_n}) + \mathbf{v}\|^2 + 2\left(r_{\boldsymbol{\theta}}^{\star}(\hat{\mathbf{x}}_{t_n}) + \mathbf{v}\right)^T \mathbf{x}_i \right]$$

$$\overset{(ii)}{=} \frac{1}{2h^2 N} \sum_i \mathcal{N}\left(\hat{\mathbf{x}}_{t_n}; \mathbf{x}_i, h^2 \mathbf{I}\right) \left[ \|\hat{\mathbf{x}}_{t_n}\|^2 - 2\hat{\mathbf{x}}_{t_n}^T r_{\boldsymbol{\theta}}^{\star}(\hat{\mathbf{x}}_{t_n}) - \|r_{\boldsymbol{\theta}}^{\star}(\hat{\mathbf{x}}_{t_n}) + \mathbf{v}\|^2 + 2\left(r_{\boldsymbol{\theta}}^{\star}(\hat{\mathbf{x}}_{t_n}) + \mathbf{v}\right)^T r_{\boldsymbol{\theta}}^{\star}(\hat{\mathbf{x}}_{t_n}) \right]$$

$$= \frac{1}{2h^2 N} \sum_i \mathcal{N}\left(\hat{\mathbf{x}}_{t_n}; \mathbf{x}_i, h^2 \mathbf{I}\right) \left[ \|\hat{\mathbf{x}}_{t_n}\|^2 - 2\hat{\mathbf{x}}_{t_n}^T r_{\boldsymbol{\theta}}^{\star}(\hat{\mathbf{x}}_{t_n}) + \|r_{\boldsymbol{\theta}}^{\star}(\hat{\mathbf{x}}_{t_n})\|^2 - \|\mathbf{v}\|^2 \right]$$

$$= \frac{1}{2h^2 N} \sum_i \mathcal{N}\left(\hat{\mathbf{x}}_{t_n}; \mathbf{x}_i, h^2 \mathbf{I}\right) \left[ \|r_{\boldsymbol{\theta}}^{\star}(\hat{\mathbf{x}}_{t_n}) - \hat{\mathbf{x}}_{t_n}\|^2 - \|\mathbf{v}\|^2 \right]$$

$$\geq 0,$$

$$\tag{39}$$

where (i) uses the definition of convex function $f(\mathbf{x}_2) \geq f(\mathbf{x}_1) + f'(\mathbf{x}_1)(\mathbf{x}_2 - \mathbf{x}_1)$ with $f(\mathbf{x}) = \exp\left(-\frac{1}{2}\mathbf{x}\right)$, $\mathbf{x}_1 = \|(\hat{\mathbf{x}}_{t_n} - \mathbf{x}_i)/h\|^2$ and $\mathbf{x}_2 = \|(r_{\boldsymbol{\theta}}^{\star}(\hat{\mathbf{x}}_{t_n}) + \mathbf{v} - \mathbf{x}_i)/h\|^2$; (ii) uses the relationship between two consecutive steps $\hat{\mathbf{x}}_{t_n}$ and $r_{\boldsymbol{\theta}}^{\star}(\hat{\mathbf{x}}_{t_n})$ in mean shift with the Gaussian kernel (see Eq. (8))

$$r_{\boldsymbol{\theta}}^{\star}(\hat{\mathbf{x}}_{t_n}) = \mathbf{m}(\hat{\mathbf{x}}_{t_n}) = \sum_i \frac{\exp\left(-\|\hat{\mathbf{x}}_{t_n} - \mathbf{x}_i\|^2 / 2h^2\right)}{\sum_j \exp\left(-\|\hat{\mathbf{x}}_{t_n} - \mathbf{x}_j\|^2 / 2h^2\right)} \mathbf{x}_i, \tag{40}$$

which implies the following equation also holds

$$\sum_i \mathcal{N}\left(\hat{\mathbf{x}}_{t_n}; \mathbf{x}_i, h^2 \mathbf{I}\right) \mathbf{x}_i = \sum_i \mathcal{N}\left(\hat{\mathbf{x}}_{t_n}; \mathbf{x}_i, h^2 \mathbf{I}\right) r_{\boldsymbol{\theta}}^{\star}(\hat{\mathbf{x}}_{t_n}). \tag{41}$$

Since $\|r_{\boldsymbol{\theta}}^{\star}(\hat{\mathbf{x}}_{t_n}) - \hat{\mathbf{x}}_{t_n}\| \geq \|\mathbf{v}\|$, or equivalently, $\|r_{\boldsymbol{\theta}}^{\star}(\hat{\mathbf{x}}_{t_n}) - \hat{\mathbf{x}}_{t_n}\|^2 \geq \|\mathbf{v}\|^2$, we conclude that the sample likelihood monotonically increases from $\hat{\mathbf{x}}_{t_n}$ to $r_{\boldsymbol{\theta}}^{\star}(\hat{\mathbf{x}}_{t_n}) + \mathbf{v}$ unless $\hat{\mathbf{x}}_{t_n} = r_{\boldsymbol{\theta}}^{\star}(\hat{\mathbf{x}}_{t_n}) + \mathbf{v}$, in terms of the kernel density estimate $p_h(\mathbf{x}) = \frac{1}{N} \sum_i \mathcal{N}(\mathbf{x}; \mathbf{x}_i, h^2 \mathbf{I})$ with any positive bandwidth $h$.

We next provide two settings for $\mathbf{v}$, which trivially satisfy the condition $\|r_{\boldsymbol{\theta}}^{\star}(\hat{\mathbf{x}}_{t_n}) - \hat{\mathbf{x}}_{t_n}\| \geq \|\mathbf{v}\|$, and have the following corollaries:

- $p_h(r_{\boldsymbol{\theta}}(\hat{\mathbf{x}}_{t_n})) \geq p_h(\hat{\mathbf{x}}_{t_n})$, when $\mathbf{v} = r_{\boldsymbol{\theta}}(\hat{\mathbf{x}}_{t_n}) - r_{\boldsymbol{\theta}}^{\star}(\hat{\mathbf{x}}_{t_n})$.

- $p_h(\hat{\mathbf{x}}_{t_{n-1}}) \geq p_h(\hat{\mathbf{x}}_{t_n})$, when $\mathbf{v} = r_{\boldsymbol{\theta}}(\hat{\mathbf{x}}_{t_n}) - r_{\boldsymbol{\theta}}^{\star}(\hat{\mathbf{x}}_{t_n}) + \frac{t_{n-1}}{t_n}\left(\hat{\mathbf{x}}_{t_n} - r_{\boldsymbol{\theta}}(\hat{\mathbf{x}}_{t_n})\right)$.

$\square$

### C.2 MAGNITUDE EXPANSION PROPERTY (PROPOSITION 2 IN THE MAIN TEXT)

**Proposition 2.** *Given a high-dimensional vector* $\mathbf{x} \in \mathbb{R}^d$ *and an isotropic Gaussian noise* $\mathbf{z} \sim \mathcal{N}(\mathbf{0}; \sigma^2 \mathbf{I}_d)$, $\sigma > 0$, *we have* $\mathbb{E} \|\mathbf{z}\|^2 = \sigma^2 d$, *and with high probability,* $\mathbf{z}$ *stays within a "thin shell":* $\|\mathbf{z}\| = \sigma\sqrt{d} \pm O(1)$. *Additionally,* $\mathbb{E} \|\mathbf{x} + \mathbf{z}\|^2 = \|\mathbf{x}\|^2 + \sigma^2 d$, $\lim_{d \to \infty} \mathbb{P}(\|\mathbf{x} + \mathbf{z}\| > \|\mathbf{x}\|) = 1$.

*Proof.* We denote $\mathbf{z}_i$ as the $i$-th dimension of random variable $\mathbf{z}$, then the expectation and variance is $\mathbb{E}[\mathbf{z}_i] = 0$, $\mathbb{V}[\mathbf{z}_i] = \sigma^2$, respectively. The fourth central moment is $\mathbb{E}[\mathbf{z}_i^4] = 3\sigma^4$. Additionally,

$$\mathbb{E}[\mathbf{z}_i^2] = \mathbb{V}[\mathbf{z}_i] + \mathbb{E}[\mathbf{z}_i]^2 = \sigma^2, \qquad \mathbb{E}[\|\mathbf{z}\|^2] = \mathbb{E}\left[\sum_{i=1}^d \mathbf{z}_i^2\right] = \sum_{i=1}^d \mathbb{E}[\mathbf{z}_i^2] = \sigma^2 d,$$

$$\mathbb{V}[\|\mathbf{z}\|^2] = \mathbb{E}[\|\mathbf{z}\|^4] - (\mathbb{E}[\|\mathbf{z}\|^2])^2 = 2d\sigma^4, \tag{42}$$

Then, we have

$$\mathbb{E}[\|\mathbf{x} + \mathbf{z}\|^2 - \|\mathbf{x}\|^2] = \mathbb{E}[\|\mathbf{z}\|^2 + 2\mathbf{x}^T \mathbf{z}] = \mathbb{E}[\|\mathbf{z}\|^2] = \sigma^2 d. \tag{43}$$

Furthermore, the standard deviation of $\|\mathbf{z}\|^2$ is $\sigma^2\sqrt{2d}$, which means

$$\|\mathbf{z}\|^2 = \sigma^2 d \pm \sigma^2\sqrt{2d} = \sigma^2 d \pm O\left(\sqrt{d}\right), \qquad \|\mathbf{z}\| = \sigma\sqrt{d} \pm O(1), \tag{44}$$

holds with high probability. In fact, we can bound the sub-gaussian tail as follows

$$\mathbb{P}\left(|\|\mathbf{z}\| - \sigma\sqrt{n}| \geq t\right) \leq 2\exp\left(-ct^2\right), \quad \forall t \geq 0, \tag{45}$$

where $c$ is an absolute constant. This property indicates that the magnitude of Gaussian random variable distributes very close to the sphere of radius $\sigma\sqrt{n}$ in the high-dimensional case.

Suppose $t = \sigma\sqrt{d} - 2\|\mathbf{x}\| > 0$, we have

$$\underbrace{\mathbb{P}(\|\mathbf{x} + \mathbf{z}\| \leq \|\mathbf{x}\|) < \mathbb{P}\left(|\|\mathbf{z}\| - \sigma\sqrt{d}| \geq \sigma\sqrt{d} - 2\|\mathbf{x}\|\right)}_{\text{See Figure 7}} \leq 2\exp\left(-c\left(\sigma\sqrt{d} - 2\|\mathbf{x}\|\right)^2\right),$$

$$\lim_{d \to \infty} \mathbb{P}(\|\mathbf{x} + \mathbf{z}\| \leq \|\mathbf{x}\|) < \lim_{d \to \infty} 2\exp\left(-c\left(\sigma\sqrt{d} - 2\|\mathbf{x}\|\right)^2\right) = 0. \tag{46}$$

Thus, $\lim_{d \to \infty} \mathbb{P}(\|\mathbf{x} + \mathbf{z}\| > \|\mathbf{x}\|) = 1$. $\qquad \square$

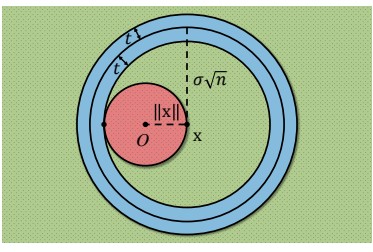

Figure 7: A simple geometric illustration (the green region is bigger than the red region).

### C.3 THE OPTIMAL DENOISING OUTPUT (PROPOSITION 4 IN THE MAIN TEXT)

Suppose we have a training dataset $\mathcal{D} = \{\mathbf{x}_i\}_{i=1}^N$ where each data $\mathbf{x}_i$ is sampled from an unknown data distribution $p_d(\mathbf{x})$, and the corresponding noisy counterpart $\hat{\mathbf{x}}_i$ is generated by a Gaussian perturbation kernel $\hat{\mathbf{x}} \sim p_t(\hat{\mathbf{x}}|\mathbf{x}) = \mathcal{N}(\hat{\mathbf{x}}; \mathbf{x}, \sigma_t^2 \mathbf{I})$ with an isotropic variance $\sigma_t^2$. The empirical data distribution $p_d(\mathbf{x})$ is denoted as a summation of multiple *Dirac delta functions* (*a.k.a* a mixture of Gaussian distribution): $p_d(\mathbf{x}) = \frac{1}{N} \sum_{i=1}^N \delta(\|\mathbf{x} - \mathbf{x}_i\|)$, and the Gaussian kernel density estimate is

$$p_t(\hat{\mathbf{x}}) = \int p_t(\hat{\mathbf{x}}|\mathbf{x}) p_d(\mathbf{x}) = \frac{1}{N} \sum_i \mathcal{N}\left(\hat{\mathbf{x}}; \mathbf{x}_i, \sigma_t^2 \mathbf{I}\right). \tag{47}$$

Based on Lemmas 1 and 3, the optimal denoising output is

$$
\begin{aligned}
r_{\boldsymbol{\theta}}^{\star}(\hat{\mathbf{x}}; \sigma_t) = \mathbb{E}(\mathbf{x}|\hat{\mathbf{x}}) &= \hat{\mathbf{x}} + \sigma_t^2 \nabla_{\hat{\mathbf{x}}} \log p_t(\hat{\mathbf{x}}) \\
&= \hat{\mathbf{x}} + \sigma_t^2 \sum_i \frac{\nabla_{\hat{\mathbf{x}}} \mathcal{N}\left(\hat{\mathbf{x}}; \mathbf{x}_i, \sigma_t^2 \mathbf{I}\right)}{\sum_j \mathcal{N}\left(\hat{\mathbf{x}}; \mathbf{x}_j, \sigma_t^2 \mathbf{I}\right)} \\
&= \hat{\mathbf{x}} + \sigma_t^2 \sum_i \frac{\mathcal{N}\left(\hat{\mathbf{x}}; \mathbf{x}_i, \sigma_t^2 \mathbf{I}\right)}{\sum_j \mathcal{N}\left(\hat{\mathbf{x}}; \mathbf{x}_j, \sigma_t^2 \mathbf{I}\right)} \left(\frac{\mathbf{x}_i - \hat{\mathbf{x}}}{\sigma_t^2}\right) \\
&= \hat{\mathbf{x}} + \sum_i \frac{\mathcal{N}\left(\hat{\mathbf{x}}; \mathbf{x}_i, \sigma_t^2 \mathbf{I}\right)}{\sum_j \mathcal{N}\left(\hat{\mathbf{x}}; \mathbf{x}_j, \sigma_t^2 \mathbf{I}\right)} (\mathbf{x}_i - \hat{\mathbf{x}}) \\
&= \sum_i \frac{\mathcal{N}\left(\hat{\mathbf{x}}; \mathbf{x}_i, \sigma_t^2 \mathbf{I}\right)}{\sum_j \mathcal{N}\left(\hat{\mathbf{x}}; \mathbf{x}_j, \sigma_t^2 \mathbf{I}\right)} \mathbf{x}_i \\
&= \sum_i \frac{\exp\left(-\|\hat{\mathbf{x}} - \mathbf{x}_i\|^2 / 2\sigma_t^2\right)}{\sum_j \exp\left(-\|\hat{\mathbf{x}} - \mathbf{x}_j\|^2 / 2\sigma_t^2\right)} \mathbf{x}_i.
\end{aligned}
\tag{48}
$$

This equation appears to be highly similar to the classic non-parametric mean shift (Fukunaga & Hostetler, 1975; Cheng, 1995; Comaniciu & Meer, 2002), especially annealed mean shift (Shen et al., 2005). The detailed discussion is provided in Section 5.

## D  RETHINKING FAST SAMPLING TECHNIQUES

The slow sampling speed is a major bottleneck of diffusion models. To address this issue, one potential solution is to replace the vanilla Euler solver with higher-order ODE solvers that use larger or adaptive step sizes (Lu et al., 2022; Zhang & Chen, 2023). This sort of training-free approaches improve the synthesis quality almost for free, but their NFEs ($\geq 10$) are still too large compared to GANs (Karras et al., 2020; Sauer et al., 2022). In fact, due to the inevitable truncation error in each numerical step (unless exactly linear trajectory), their minimum achievable NFE is limited by the curvature of ODE trajectory. Another solution is to explicitly straighten the ODE trajectory with knowledge distillation (Luhman & Luhman, 2021; Salimans & Ho, 2022; Zheng et al., 2023; Song et al., 2023). They optimize a new student score model ($r_{\boldsymbol{\theta}}$) to align with the prediction of a pre-trained and fixed teacher score model ($r_{\boldsymbol{\phi}}$). Empirically, these learning-based approaches can achieve decent synthesis quality with a significantly fewer NFE ($\approx 1$) compared to those ODE solver-based approaches.

**Rethinking Distillation-Based Techniques.** A learned score-based model with a specified ODE solver fully determine the sampling trajectory and the denoising trajectory. Various distillation-based fast sampling techniques can be interpreted as different ways to *linearize* the original trajectory. We provide a sketch in Figure 8 to highlight the difference of typical examples, including KD (Luhman & Luhman, 2021), PD (Salimans & Ho, 2022), DFNO (Zheng et al., 2023), and CD (Song et al., 2023). We can see that the student sampling direction is adjusted to make the initial sample directly point to the synthetic sample of the pre-defined teacher model. To achieve this goal, new noise-target pairs are built in an online (Salimans & Ho, 2022; Song et al., 2023; Berthelot et al., 2023) or offline fashion (Luhman & Luhman, 2021; Zheng et al., 2023). Recently, CD (Song et al., 2023) and TRACT (Berthelot et al., 2023) began to rely on the denoising trajectory to guide the score fine-tuning and thus enable fast sampling. We present the training objective of CD as follows

$$
\mathbb{E}_{\mathbf{x} \sim p_d} \mathbb{E}_{\mathbf{z} \sim \mathcal{N}(\mathbf{0}, t_{n+1}^2 \mathbf{I})} \| r_{\boldsymbol{\theta}}(\hat{\mathbf{x}}; t_{n+1}) - r_{\boldsymbol{\theta}^-}(\mathcal{T}_{\boldsymbol{\phi}}(\hat{\mathbf{x}}); t_n) \|_2^2, \quad \hat{\mathbf{x}} = \mathbf{x} + \mathbf{z},
\tag{49}
$$

where $\mathcal{T}_{\boldsymbol{\phi}}(\hat{\mathbf{x}})$ is implemented as a one-step Euler: $\frac{t_n}{t_{n+1}} \hat{\mathbf{x}} + \frac{t_{n+1} - t_n}{t_{n+1}} r_{\boldsymbol{\phi}}(\hat{\mathbf{x}}; t_{n+1})$. CD follows the time schedule of EDMs (Karras et al., 2022), aside from removing the $t_0$, and adopts pre-trained EDMs to initialize the student model. $\boldsymbol{\theta}^-$ is the exponential moving average (EMA) of $\boldsymbol{\theta}$.

Based on our geometric observations, we then provide an intuitive interpretation of the training objective Eq. (49), as shown in Figure 8d. (1) The role of $\mathcal{T}_{\boldsymbol{\phi}}(\hat{\mathbf{x}})$ is to locate the sampling trajectory passing a given noisy sample $\hat{\mathbf{x}}$ and make one numerical step along the trajectory to move $\hat{\mathbf{x}}$ towards its converged point. (2) $r_{\boldsymbol{\theta}^-}(\cdot)$ further projects $\mathcal{T}_{\boldsymbol{\phi}}(\hat{\mathbf{x}})$ into the corresponding denoising trajectory

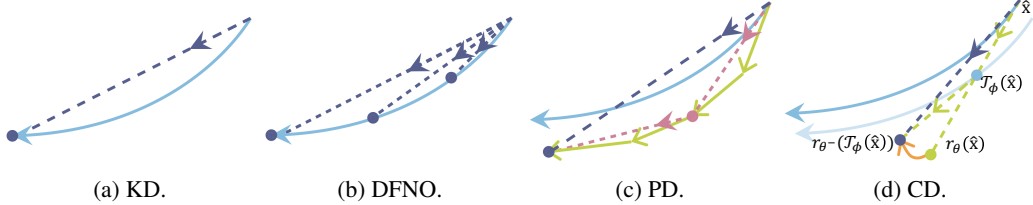

(a) KD.                (b) DFNO.                (c) PD.                (d) CD.

Figure 8: The comparison of distillation-based techniques. The *offline* techniques first simulate a long ODE trajectory with the teacher score and then make the student score point to the final sample (KD (Luhman & Luhman, 2021)) or also intermediate samples on the trajectory (DFNO (Zheng et al., 2023)). The *online* techniques iteratively fine-tune the student prediction to align with the target generated by the few-step teacher model along the sampling trajectory (PD (Salimans & Ho, 2022)) or the denoising trajectory (CD (Song et al., 2023)).

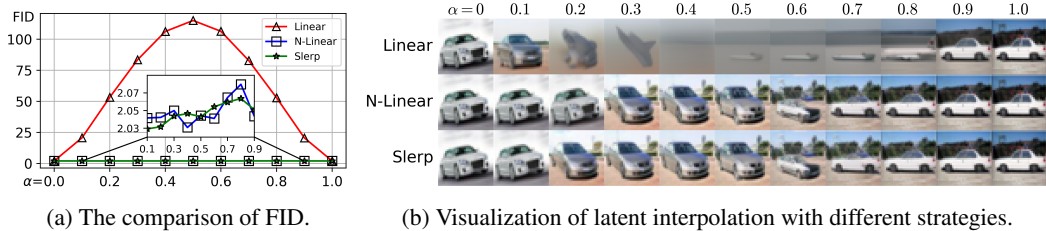

(a) The comparison of FID.        (b) Visualization of latent interpolation with different strategies.

Figure 9: Linear latent interpolation results in blurry images, while a simple re-scaling trick greatly preserves the fine-grained image details and enables a smooth traversal among different modes.

with the step size $t_n$, which is closer to the converged point compared with $r_{\boldsymbol{\theta}}(\hat{\mathbf{x}})$ . (3) The student denoising output $r_{\boldsymbol{\theta}}(\cdot)$ is then shifted to match its underlying target $r_{\boldsymbol{\theta}^-}(\cdot)$ in the denoising trajectory. By iteratively fine-tuning denoising outputs until convergence, the student model is hopefully endowed with the ability to perform few-step sampling from those trained discrete time steps, and thus achieves excellent performance in practice (Song et al., 2023).

## E    IN-DISTRIBUTION LATENT INTERPOLATION

An attractive application of diffusion models is to achieve semantic image editing by manipulating latent representations (Ho et al., 2020; Song et al., 2021a;c). Such semantic editing is relatively controllable for ODE-based sampling due to its deterministic trajectory, compared with SDE-based sampling that injects stochastic noise in each step. We then take *latent interpolation* as an example to reveal the potential pitfalls in practice from a geometric viewpoint.

The training objective Eq. (3) for score estimation tells that given a noise level $\sigma_t^2$, the denoiser is *only* trained with samples *belonging to* the distribution $p_t(\hat{\mathbf{x}})$. This important fact implies that for the latent encoding $\hat{\mathbf{x}}_{t_N} \sim \mathcal{N}(\mathbf{0}, T^2\mathbf{I})$, the denoiser performance is only guaranteed for the input approximately distributed in a sphere of radius $T\sqrt{d}$ (see Section 3.1). This geometric picture helps in understanding the conditions under which latent interpolation may fail.

**Proposition 8.** *In high dimensions, linear interpolation shifts the latent distribution while spherical linear interpolation asymptotically ($d \to \infty$) maintains the latent distribution.*

Given two independent latent encodings, $\hat{\mathbf{x}}_{t_N}^{(1)}, \hat{\mathbf{x}}_{t_N}^{(2)} \sim \mathcal{N}(\mathbf{0}, T^2\mathbf{I})$, they are almost orthogonal with the angle $\frac{1}{2}\pi + O_p(d^{-0.5})$ in high dimensions (Hall et al., 2005; Vershynin, 2018). In this case, *linear interpolation* $\hat{\mathbf{x}}_{t_N}^{(\alpha)} = (1 - \alpha)\hat{\mathbf{x}}_{t_N}^{(1)} + \alpha\hat{\mathbf{x}}_{t_N}^{(2)}$ (Ho et al., 2020) quickly pushes the resulting encoding $\hat{\mathbf{x}}_{t_N}^{(\alpha)}$ away from the original distribution into a squashed sphere of radius $T\sqrt{d((1-\alpha)^2+\alpha^2)}$, which almost has no intersection with the original sphere of radius $T\sqrt{d}$, unless $\alpha \to 0$ or 1. Our trained denoiser thus can not provide a reliable estimation for $r_{\boldsymbol{\theta}}(\hat{\mathbf{x}}_{t_N}^{(\alpha)}, t_N)$ to derive the score direction, as shown in Figure 9. Another strategy named as *spherical linear interpolation* (slerp) (Song et al.,

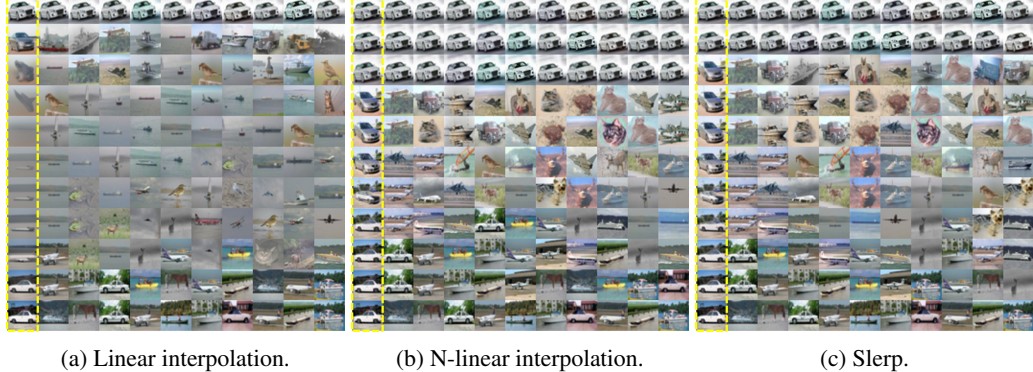

(a) Linear interpolation.    (b) N-linear interpolation.    (c) Slerp.

Figure 10: We provide the k-nearest neighbors (k=10) of our generated images to the real data, in order to demonstrate how different modes are smoothly traversed in the latent interpolation process.

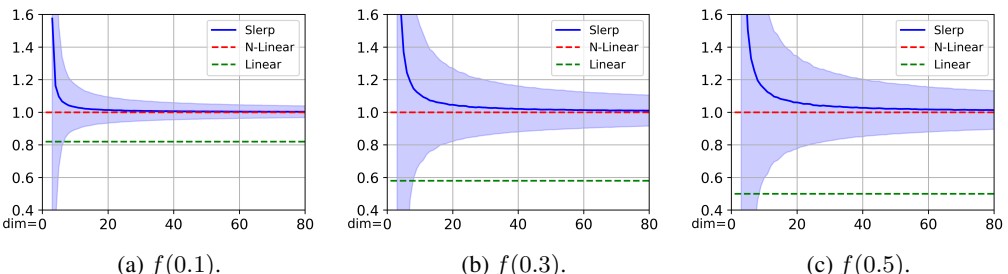

(a) $f(0.1)$.    (b) $f(0.3)$.    (c) $f(0.5)$.

Figure 11: The comparison of three interpolation strategies with different dimensions.

2021a;c; Ramesh et al., 2022; Song et al., 2023) greatly alleviates (but is not free from) the squashing effect in high dimensions and thus stabilizes the synthesis quality of interpolated encodings. But it still suffers from distribution shift in low dimensional cases (see Section E.1). The comparison of above two strategies further brings out a basic concept called *in-distribution interpolation*.

**Proposition 9.** *In-distribution interpolation preserves the latent distribution under interpolation.*

This concept gives rise to a family of interpolation strategies. In particular, for the Gaussian encoding $\hat{\mathbf{x}}_{t_N}$, there exists a variance-preserving interpolation $\hat{\mathbf{x}}_{t_N}^{(\lambda)} = \sqrt{(1-\lambda^2)}\hat{\mathbf{x}}_{t_N}^{(1)} + \lambda\hat{\mathbf{x}}_{t_N}^{(2)} \sim \mathcal{N}\left(\mathbf{0}, T^2\mathbf{I}\right)$ to prevent distribution shift. Since a uniform $\lambda$ makes $\hat{\mathbf{x}}_{t_N}^{(\lambda)}$ largely biased to $\hat{\mathbf{x}}_{t_N}^{(1)}$, we derive $\lambda$ by re-scaling other heuristic strategies to scatter the coefficient more evenly, such as the *normalized linear* (n-linear) interpolation ($\lambda = \alpha/\sqrt{\alpha^2 + (1-\alpha^2)}$) with uniformly sampled coefficient $\alpha$. As shown in Figure 9, this simple re-scaling trick significantly boosts the visual quality compared with the original counterpart. Additionally, slerp behaves as $\lambda = \sin\alpha\frac{\pi}{2}$ in high dimensions due to $\psi \approx \frac{\pi}{2}$, and this coefficient was used in (Dhariwal & Nichol, 2021) for interpolation.

With the help of such an in-distribution interpolation, all interpolated encodings faithfully move along our trained ODE trajectory with a reliable denoising estimation for $r_{\boldsymbol{\theta}}(\hat{\mathbf{x}}_{t_N}^{(\lambda)}, t_N)$. We further calculate the k-nearest neighbors of our generated images to the real data (see Figure 10), to demonstrate how different modes are smoothly traversed in this process $\hat{\mathbf{x}}_{t_N} \to \hat{\mathbf{x}}_{t_N}^\lambda \to \hat{\mathbf{x}}_{t_0}^\lambda$.

### E.1 DETAILED COMPARISON OF THREE LATENT INTERPOLATION STRATEGIES

Given two independent latent encodings, $\hat{\mathbf{x}}_{t_N}^{(1)}, \hat{\mathbf{x}}_{t_N}^{(2)} \sim \mathcal{N}(\mathbf{0}, T^2\mathbf{I})$, and $0 < \alpha < 1$, we have three latent interpolation strategies as follows and their comparison is summarized in Table 2.

- *linear interpolation*:

$$\hat{\mathbf{x}}_{t_N}^{(\alpha)} = (1-\alpha)\hat{\mathbf{x}}_{t_N}^{(1)} + \alpha\hat{\mathbf{x}}_{t_N}^{(2)}; \tag{50}$$

- *spherical linear interpolation* (slerp):

$$\hat{\mathbf{x}}_{t_N}^{(\alpha)} = \frac{\sin[(1-\alpha)\psi]}{\sin(\psi)}\hat{\mathbf{x}}_{t_N}^{(1)} + \frac{\sin(\alpha\psi)}{\sin(\psi)}\hat{\mathbf{x}}_{t_N}^{(2)}, \quad \psi = \arccos\left(\frac{\langle\hat{\mathbf{x}}_{t_N}^{(1)}, \hat{\mathbf{x}}_{t_N}^{(2)}\rangle}{\|\hat{\mathbf{x}}_{t_N}^{(1)}\|_2 \cdot \|\hat{\mathbf{x}}_{t_N}^{(2)}\|_2}\right), \quad (51)$$

  where $\psi$ is calculated as the angle subtended by the arc connecting two encodings;
- *in-distribution interpolation*:

$$\hat{\mathbf{x}}_{t_N}^{(\lambda)} = \sqrt{(1-\lambda^2)}\hat{\mathbf{x}}_{t_N}^{(1)} + \lambda\hat{\mathbf{x}}_{t_N}^{(2)}. \quad (52)$$

  In particular, by setting $\lambda = \alpha/\sqrt{\alpha^2 + (1-\alpha^2)}$, we obtain
- *normalized linear interpolation* (n-linear):

$$\hat{\mathbf{x}}_{t_N}^{(\alpha)} = \frac{\sqrt{1-\alpha^2}}{\sqrt{\alpha^2 + (1-\alpha^2)}}\hat{\mathbf{x}}_{t_N}^{(1)} + \frac{\alpha}{\sqrt{\alpha^2 + (1-\alpha^2)}}\hat{\mathbf{x}}_{t_N}^{(2)}. \quad (53)$$

Thanks to the linearity of Gaussian random variable, the interpolated latent encoding by each of the above strategies still belongs to a Gaussian distribution with zero mean. We thus only need to check the variance to see whether the distribution shift happens.

The variance comparison is provided as follows

- *linear interpolation*:

$$\begin{aligned}
\mathbb{V}\left(\hat{\mathbf{x}}_{t_N}^{(\alpha)}\right) &= \mathbb{V}\left((1-\alpha)\hat{\mathbf{x}}_{t_N}^{(1)} + \alpha\hat{\mathbf{x}}_{t_N}^{(2)}\right) \\
&= (1-\alpha)^2\mathbb{V}\left(\hat{\mathbf{x}}_{t_N}^{(1)}\right) + \alpha^2\mathbb{V}\left(\hat{\mathbf{x}}_{t_N}^{(2)}\right) \\
&= \left((1-\alpha)^2 + \alpha^2\right)T^2\mathbf{I} \stackrel{\text{def}}{=} f(\alpha)T^2\mathbf{I},
\end{aligned} \quad (54)$$

  where $\sup f(\alpha) = 1$ if $\alpha \to 0$ or $1$, and $\min f(\alpha) = 0.5$ if $\alpha = 0.5$.
- *spherical linear interpolation* (slerp):

$$\begin{aligned}
\mathbb{V}\left(\hat{\mathbf{x}}_{t_N}^{(\alpha)}\right) &= \mathbb{V}\left(\frac{\sin[(1-\alpha)\psi]}{\sin(\psi)}\hat{\mathbf{x}}_{t_N}^{(1)} + \frac{\sin(\alpha\psi)}{\sin(\psi)}\hat{\mathbf{x}}_{t_N}^{(2)}\right) \\
&= \frac{\sin^2[(1-\alpha)\psi] + \sin^2(\alpha\psi)}{\sin^2(\psi)}T^2\mathbf{I} \stackrel{\text{def}}{=} f(\alpha)T^2\mathbf{I},
\end{aligned} \quad (55)$$

  where $f(0.5) = \frac{2\sin^2[\psi/2]}{\sin^2(\psi)} = \frac{1}{2\cos^2(\psi/2)} = 2\sec^2(\psi/2) \geq 0.5$.

  Since these two latent encodings $\hat{\mathbf{x}}_{t_N}^{(1)}$ and $\hat{\mathbf{x}}_{t_N}^{(2)}$ are almost orthogonal with the angle $\frac{1}{2}\pi + O_p(d^{-0.5})$ in high dimensions (Hall et al., 2005; Vershynin, 2018), we can conclude that asymptotically,

$$\lim_{d\to\infty}\mathbb{V}\left(\hat{\mathbf{x}}_{t_N}^{(\alpha)}\right) = \left(\sin^2[(1-\alpha)\frac{\pi}{2}] + \sin^2(\alpha\frac{\pi}{2})\right)T^2\mathbf{I} = T^2\mathbf{I}, \quad (56)$$

- *in-distribution interpolation*:

$$\mathbb{V}\left(\hat{\mathbf{x}}_{t_N}^{(\lambda)}\right) = \mathbb{V}\left(\sqrt{(1-\lambda^2)}\hat{\mathbf{x}}_{t_N}^{(1)} + \lambda\hat{\mathbf{x}}_{t_N}^{(2)}\right) = f(\lambda)T^2\mathbf{I} = T^2\mathbf{I}. \quad (57)$$

We provide a comparison of three interpolation strategies with different dimensions in Figure 11.

**Remark 5.** *In high dimensions, linear interpolation (Ho et al., 2020) shifts the latent distribution while spherical linear interpolation (Song et al., 2021a) asymptotically ($d \to \infty$) maintains the latent distribution.*

**Remark 6.** *In-distribution interpolation preserves the latent distribution under interpolation.*

**Remark 7.** *If two latent encodings share the same magnitude, i.e., $\|\hat{\mathbf{x}}_{t_N}^{(1)}\|^2 = \|\hat{\mathbf{x}}_{t_N}^{(2)}\|^2 = C^2$, then spherical linear interpolation maintains the magnitude, i.e., $\|\hat{\mathbf{x}}_{t_N}^{(\alpha)}\|^2 = C^2$.*

Table 2: The comparison of three latent interpolation strategies.

| Interpolation | Distribution Shift | Magnitude Shift |
|---|---|---|
| Linear | ✓ | ✓ |
| Spherical linear | (asymptotic) ✗ | ✗ |
| In-distribution | ✗ | (asymptotic, approximate) ✗ |

*Proof.* Since $\sin\left[(1-\alpha)\,\psi\right] = \sin\psi\cos(\alpha\psi) - \cos\psi\sin(\alpha\psi)$, we have

$$
\begin{aligned}
\|\hat{\mathbf{x}}_{t_N}^{(\alpha)}\|^2 &= \left\|\frac{\sin\left[(1-\alpha)\,\psi\right]}{\sin\psi}\hat{\mathbf{x}}_{t_N}^{(1)} + \frac{\sin(\alpha\psi)}{\sin\psi}\hat{\mathbf{x}}_{t_N}^{(2)}\right\|^2 \\
&= \frac{1}{\sin^2\psi}\left(\sin^2\left[(1-a)\,\psi\right]\|\hat{\mathbf{x}}_{t_N}^{(1)}\|^2 + \sin^2(\alpha\psi)\|\hat{\mathbf{x}}_{t_N}^{(2)}\|^2\right. \\
&\quad \left. +2\sin\left[(1-\alpha)\,\psi\right]\sin(\alpha\psi)\langle\hat{\mathbf{x}}_{t_N}^{(1)},\hat{\mathbf{x}}_{t_N}^{(2)}\rangle\right),
\end{aligned}
\tag{58}
$$

where

$$
\begin{aligned}
&\sin^2\left[(1-a)\,\psi\right]\|\hat{\mathbf{x}}_{t_N}^{(1)}\|^2 + \sin^2(\alpha\psi)\|\hat{\mathbf{x}}_{t_N}^{(2)}\|^2 \\
&= \left\{\left[\sin^2\psi\cos^2(\alpha\psi) + \cos^2\psi\sin^2(\alpha\psi) - 2\sin\psi\cos\psi\sin\alpha\psi\cos\alpha\psi\right] + \sin^2(\alpha\psi)\right\}C^2 \\
&= \left[\sin^2\psi + 2\cos^2\psi\sin^2(\alpha\psi) - 2\sin\psi\cos\psi\sin(\alpha\psi)\cos(\alpha\psi)\right]C^2
\end{aligned}
\tag{59}
$$

and

$$
\begin{aligned}
&2\sin\left[(1-\alpha)\,\psi\right]\sin(\alpha\psi)\langle\hat{\mathbf{x}}_{t_N}^{(1)},\hat{\mathbf{x}}_{t_N}^{(2)}\rangle \\
&= \left[2\sin\psi\sin(\alpha\psi)\cos(\alpha\psi) - 2\cos\psi\sin^2(\alpha\psi)\right]C^2\cos\psi \\
&= \left[2\sin\psi\cos\psi\sin(\alpha\psi)\cos(\alpha\psi) - 2\cos^2\psi\sin^2(\alpha\psi)\right]C^2.
\end{aligned}
\tag{60}
$$

We then substitute Eq. (59) and Eq. (60) into Eq. (58) and obtain $\|\hat{\mathbf{x}}_{t_N}^{(\alpha)}\|^2 = C^2$. □

**Remark 8.** *If two latent encodings share the same magnitude, i.e., $\|\hat{\mathbf{x}}_{t_N}^{(1)}\|^2 = \|\hat{\mathbf{x}}_{t_N}^{(2)}\|^2 = C^2$, then linear interpolation shrinks the magnitude, i.e., $\|\hat{\mathbf{x}}_{t_N}^{(\alpha)}\|^2 < C^2$ while in-distribution interpolation approximately preserves the magnitude in high dimensions.*

*Proof.* In high dimensions ($d \gg 1$, $C = T\sqrt{d} \gg 1$), $|\langle\hat{\mathbf{x}}_{t_N}^{(1)},\hat{\mathbf{x}}_{t_N}^{(2)}\rangle| \approx C$ (Vershynin, 2018). Then, we have

- *linear interpolation:*

$$
\begin{aligned}
\|\hat{\mathbf{x}}_{t_N}^{(\alpha)}\|^2 &= \left\|(1-\alpha)\hat{\mathbf{x}}_{t_N}^{(1)} + \alpha\hat{\mathbf{x}}_{t_N}^{(2)}\right\|^2 \\
&= (1-\alpha)^2\|\hat{\mathbf{x}}_{t_N}^{(1)}\|^2 + \alpha^2\|\hat{\mathbf{x}}_{t_N}^{(2)}\|^2 + 2\alpha(1-\alpha)\langle\hat{\mathbf{x}}_{t_N}^{(1)},\hat{\mathbf{x}}_{t_N}^{(2)}\rangle \\
&\approx \left((1-\alpha)^2 + \alpha^2\right)C^2 < C^2.
\end{aligned}
\tag{61}
$$

- *in-distribution interpolation:*

$$
\begin{aligned}
\|\hat{\mathbf{x}}_{t_N}^{(\alpha)}\|^2 &= \left\|\sqrt{(1-\lambda^2)}\hat{\mathbf{x}}_{t_N}^{(1)} + \lambda\hat{\mathbf{x}}_{t_N}^{(2)}\right\|^2 \\
&= (1-\lambda^2)\|\hat{\mathbf{x}}_{t_N}^{(1)}\|^2 + \lambda^2\|\hat{\mathbf{x}}_{t_N}^{(2)}\|^2 + 2\lambda\sqrt{(1-\lambda^2)}\langle\hat{\mathbf{x}}_{t_N}^{(1)},\hat{\mathbf{x}}_{t_N}^{(2)}\rangle \\
&\approx C^2.
\end{aligned}
\tag{62}
$$

□

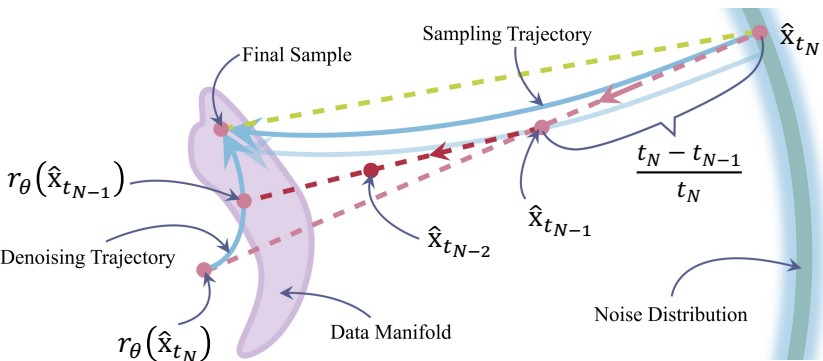

Figure 12: An illustration of two consecutive Euler steps. Each Euler step in ODE-based sampling is a convex combination of the denoising output and the current position to determine the next position.

## F ADDITIONAL EXPERIMENTAL DETAILS

### F.1 EXPERIMENTAL SETTINGS

Unless otherwise specified, we follow the configurations and experimental settings of a recent framework called EDMs, with $\mathbf{f}(\mathbf{x}, t) = \mathbf{0}$ and $g(t) = \sqrt{2t}$ (Karras et al., 2022; Song et al., 2023). In this case, the forward VE SDE is $d\mathbf{x} = \sqrt{2t}\, d\mathbf{w}_t$, and the empirical probability flow ODE is $d\mathbf{x} = \frac{\mathbf{x} - r_\theta(\mathbf{x};t)}{t}\, dt$. The default ODE-based sampler is Heun's 2$^{\text{nd}}$ order method starting from $\hat{\mathbf{x}}_{t_N} \sim \mathcal{N}(\mathbf{0}, T^2\mathbf{I})$ to $\hat{\mathbf{x}}_{t_0}$. The time horizon is divided with the formula $t_n = (t_1^{1/\rho} + \frac{n-1}{N-1}(t_N^{1/\rho} - t_1^{1/\rho}))^\rho$, where $t_1 = 0.002$, $t_N = 80$, $n \in [1, N]$ and $t_0 = 0$, $\rho = 7$ (Karras et al., 2022).

- We adopt the official implementation and pre-trained checkpoints[3] of EDMs (Karras et al., 2022) for experiments on CIFAR-10[4] (Krizhevsky & Hinton, 2009) (32×32) with $N = 18$, NFE = 35; and FFHQ[5] (Karras et al., 2019) (64×64) with $N = 40$, NFE = 79; and AFHQv2[6] (Choi et al., 2020) (64×64) with $N = 40$, NFE = 79.

- We adopt the re-implementation of EDMs and pre-trained checkpoints[7] from consistency models (Song et al., 2023) for experiments on LSUN Bedroom, LSUN Cat[8] (Yu et al., 2015) (256×256), and ImageNet-64[9] (Russakovsky et al., 2015) (64×64), with $N = 40$, NFE = 79.

The experimental results provided in Sections F.2 and F.3 confirm that the geometric structures observed in the main submission widely exist on other datasets (such as LSUN Bedroom, LSUN Cat) and other model settings (such as conditional generation, various network architectures). We also provide the preliminary results on VP-SDE in Section F.4.

In Section F.5, we demonstrate that our proposed ODE-Jump works well on other datasets (such as FFHQ, AFHQv2), other ODE-based samplers (such as DPM-Solver, PLMS), a SDE-based sampler and a popular large-scale text-to-image model called Stable Diffusion[10] (Rombach et al., 2022).

All experiments are conducted on 4 NVIDIA A100 GPUs.

---

[3] https://nvlabs-fi-cdn.nvidia.com/edm (Creative Commons Attribution-NonCommercial ShareAlike 4.0 International License)
[4] https://www.cs.toronto.edu/~kriz/cifar.html
[5] https://github.com/NVlabs/ffhq-dataset
[6] https://github.com/clovaai/stargan-v2/blob/master/README.md#animal-faces-hq-dataset-afhq
[7] https://github.com/openai/consistency_models (MIT License)
[8] https://www.yf.io/p/lsun
[9] https://www.image-net.org/index.php
[10] https://github.com/CompVis/stable-diffusion/ (CreativeML Open RAIL-M License)

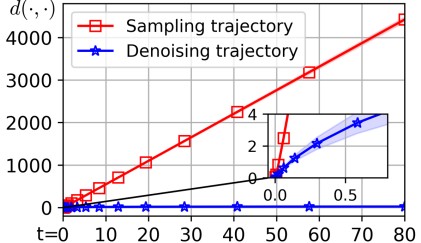

Figure 13: We compare the blue curves in Figure 2b, and find that the expected $\ell_2$ distance between the intermediate samples and the final sample in the denoising trajectory is always smaller than that of the sampling trajectory, which supports the fast convergence of the denoising trajectory.

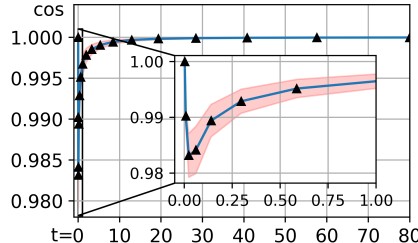

Figure 14: The angle deviation in the sampling trajectory is calculated by the cosine between the backward ODE direction $-\frac{\mathrm{d}\mathbf{x}_t}{\mathrm{d}t}\big|_t$ and the direction pointing to the final point $(\hat{\mathbf{x}}_{t_0} - \hat{\mathbf{x}}_t)$ at discrete time $t$. The angle deviation always stays in a narrow range (from about 0.98 to 1.00).

### F.2 MORE RESULTS ON THE VE-SDE

In the main submission, we conduct unconditional generation with the checkpoint[11] of the DDPM++ architecture to observe geometric structures of the sampling trajectory and the denoising trajectory. Under this experimental setup, we provide more results in this section, including:

- *A Complement of the Observation 3:* The denoising trajectory converges faster than the sampling trajectory in terms of the expected $\ell_2$ distance, as shown in Figure 13.

- *Angle Deviation:* The results are provided in Figure 14;

- *Diagnosis of Score Deviation:* Apart from Figure 5 in the main submission, we also observe that there exists a few cases where $\{r_{\boldsymbol{\theta}}^\star(\hat{\mathbf{x}}_t)\}$ and $\{r_{\boldsymbol{\theta}}^\star(\hat{\mathbf{x}}_t^\star)\}$ almost share the same final sample, as shown in Figure 15. In this case, the score deviation is relatively small such that for the final sample of the denoising trajectory, its nearest neighbor to the real image in the dataset is exactly the final sample of the corresponding optimal trajectory.

- *Trajectory Shape on Other Datasets:* The results are provided in Figure 16 and 17. Similar to Figure 2a, we track the magnitude of original data in the forward SDE process at 500 time steps uniformly distributed in $[0, 80]$ and we track the magnitude of synthesized samples in the backward ODE process, starting from 50k randomly-sampled noise. We calculate the means and standard deviations at each time step, and plot them as circle mark and short black line, respectively, in Figure 16a and 17a. We find that the magnitude shift happens in the last few time steps (see Figure 16a and 17a), unlike the case in Figure 2a where the two distributions are highly aligned. In Figure 16b and 17b, we visualize the deviation in the sampling trajectory and the denoising trajectory as Figure 2b. A sampling trajectory and its associated denoising trajectory are also provided for illustration.

---

[11]https://nvlabs-fi-cdn.nvidia.com/edm/pretrained/edm-cifar10-32x32-uncond-vp.pkl

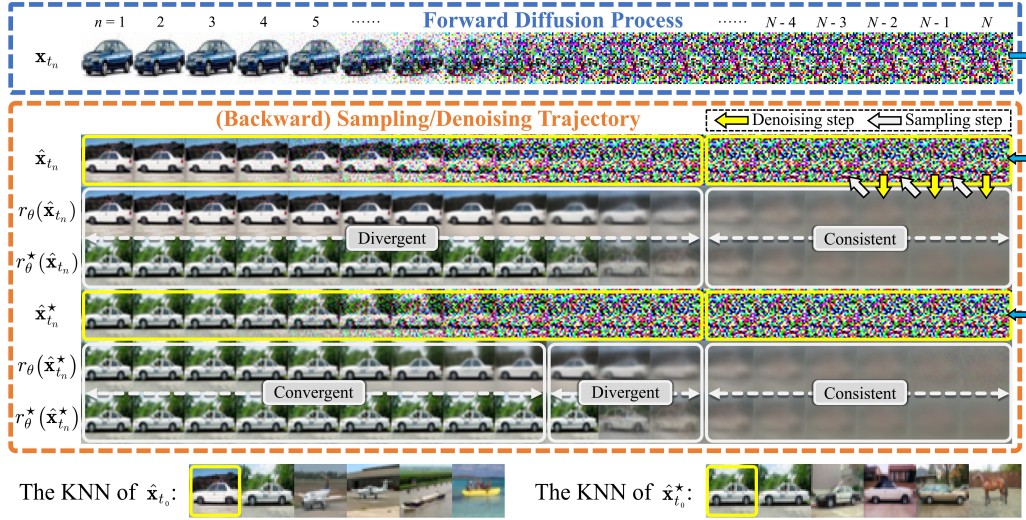

Figure 15: The unconditional generation on CIFAR-10. *Top:* We visualize a forward diffusion process of a randomly-selected image to obtain its encoding $\hat{\mathbf{x}}_{t_N}$ and simulate multiple trajectories starting from this encoding. *Bottom:* The k-nearest neighbor (k=5) of $\hat{\mathbf{x}}_{t_0}$ and $\hat{\mathbf{x}}_{t_0}^{\star}$.

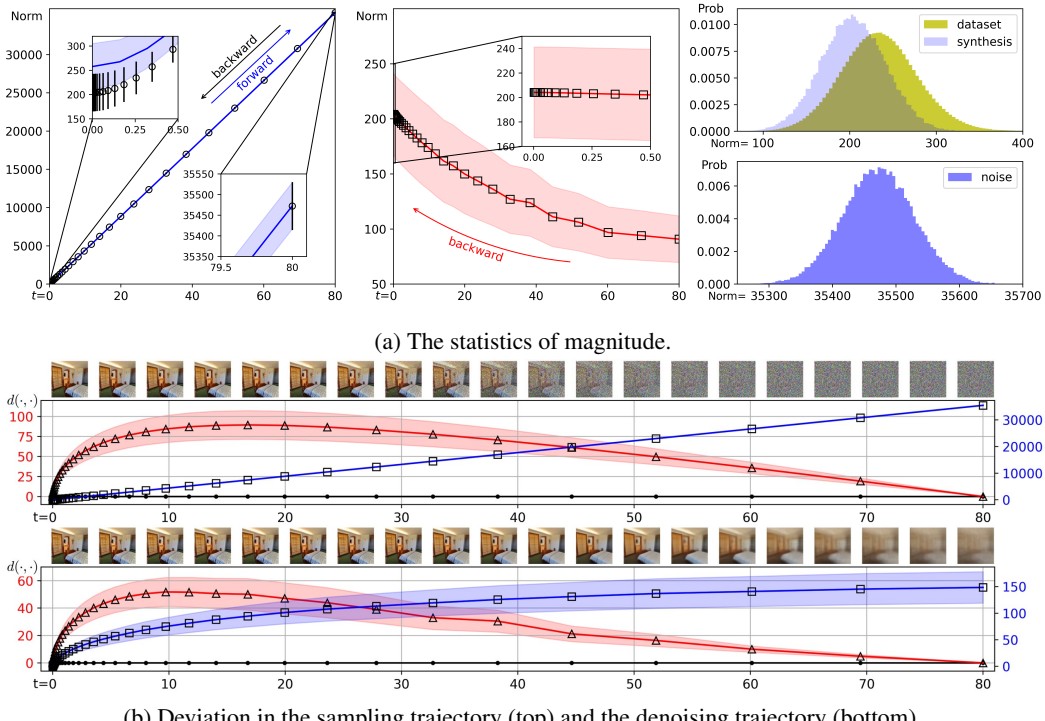

(a) The statistics of magnitude.

(b) Deviation in the sampling trajectory (top) and the denoising trajectory (bottom).

Figure 16: LSUN Bedroom.

## F.3 MORE RESULTS ON OTHER SDES

In this section, we provide more results with different VE SDE settings on CIFAR-10, including:

- The conditional generation with the checkpoint[12] of DDPM++ architecture:

---

[12] https://nvlabs-fi-cdn.nvidia.com/edm/pretrained/edm-cifar10-32x32-cond-vp.pkl

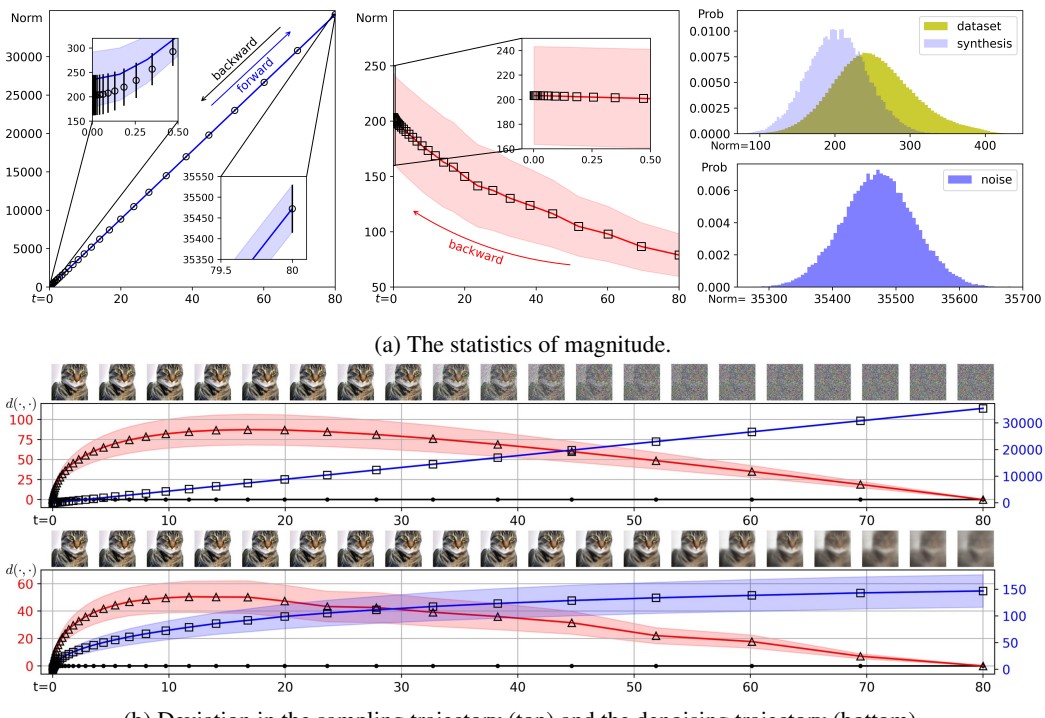

(a) The statistics of magnitude.

(b) Deviation in the sampling trajectory (top) and the denoising trajectory (bottom).

Figure 17: LSUN Cat.

- *Trajectory Shape:* The results are provided in Figure 18.
- *Diagnosis of Score Deviation:* The results are provided in Figure 19.

All observations are similar to those of the unconditional case in the main submission.

- The unconditional generation with the checkpoint[13] of NCSN++ architecture:
  - *Trajectory Shape:* The results are provided in Figure 20. The observations are similar to those of the unconditional generation case in the main submission. An outlier appears at about $t = 40$ and we find that its score deviation becomes abnormally large. This implies the model probably fails to learn a good score direction at this point.

## F.4  MORE RESULTS ON THE VP-SDE

In this section, we provide preliminary results with a variance-preserving SDE, following the configurations of EDM (Karras et al., 2022) (see Table 1 in their paper). We conduct unconditional generation with the checkpoint[14] of DDPM++ architecture.

We find that in this case, the data distribution and the noise distribution are still smoothly connected with an explicit, quasi-linear sampling trajectory, and another implicit denoising trajectory that converges faster. The geometric properties of VE-SDEs discussed in Section 3.2 are generally (but not strictly) hold for this VP-SDE.

In fact, as we discussed in Appendix A.2, we can transform other model types (*e.g.*, VP-SDEs) into the VE counterparts with change-of-variables formula and focus on the geometric behaviors of VE-SDE instead. Nevertheless, further investigation is required in order to thoroughly understand the behavior of VP-SDEs. We leave it for future work.

---

[13]https://nvlabs-fi-cdn.nvidia.com/edm/pretrained/edm-cifar10-32x32-uncond-ve.pkl

[14]https://nvlabs-fi-cdn.nvidia.com/edm/pretrained/baseline/baseline-cifar10-32x32-uncond-vp.pkl

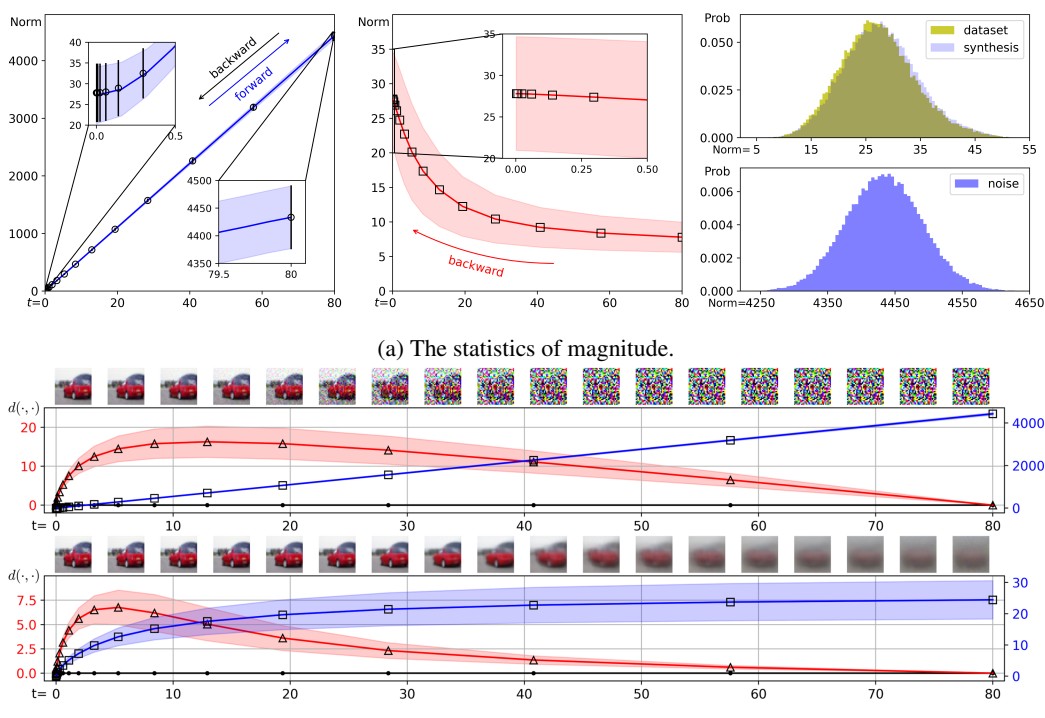

(a) The statistics of magnitude.

(b) Deviation in the sampling trajectory (top) and the denoising trajectory (bottom).

Figure 18: The conditional generation with VE-SDE and DDPM++ architecture.

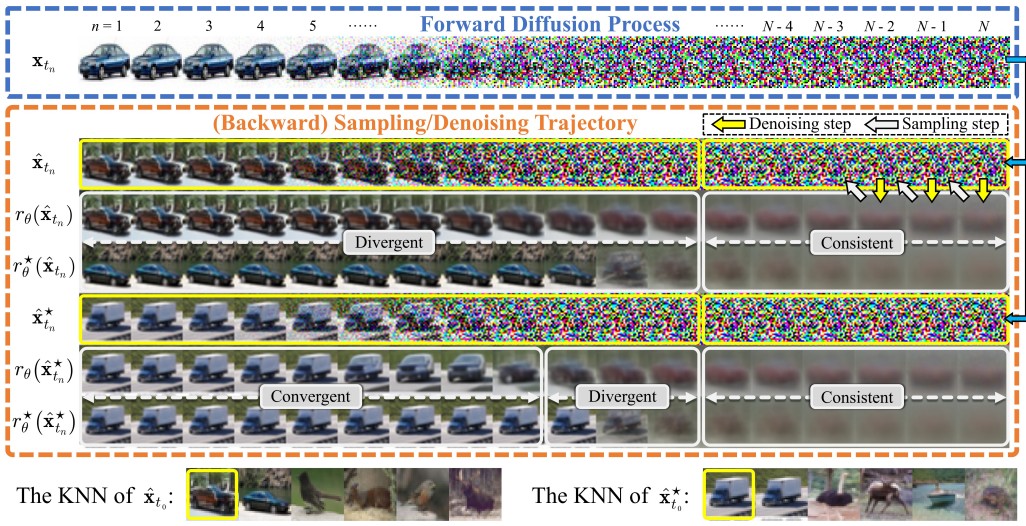

Figure 19: The conditional generation on CIFAR-10. *Top:* We visualize a forward diffusion process of a randomly-selected image to obtain its encoding $\hat{\mathbf{x}}_{t_N}$ and simulate multiple trajectories starting from this encoding. *Bottom:* The k-nearest neighbor (k=5) of $\hat{\mathbf{x}}_{t_0}$ and $\hat{\mathbf{x}}_{t_0}^{\star}$.

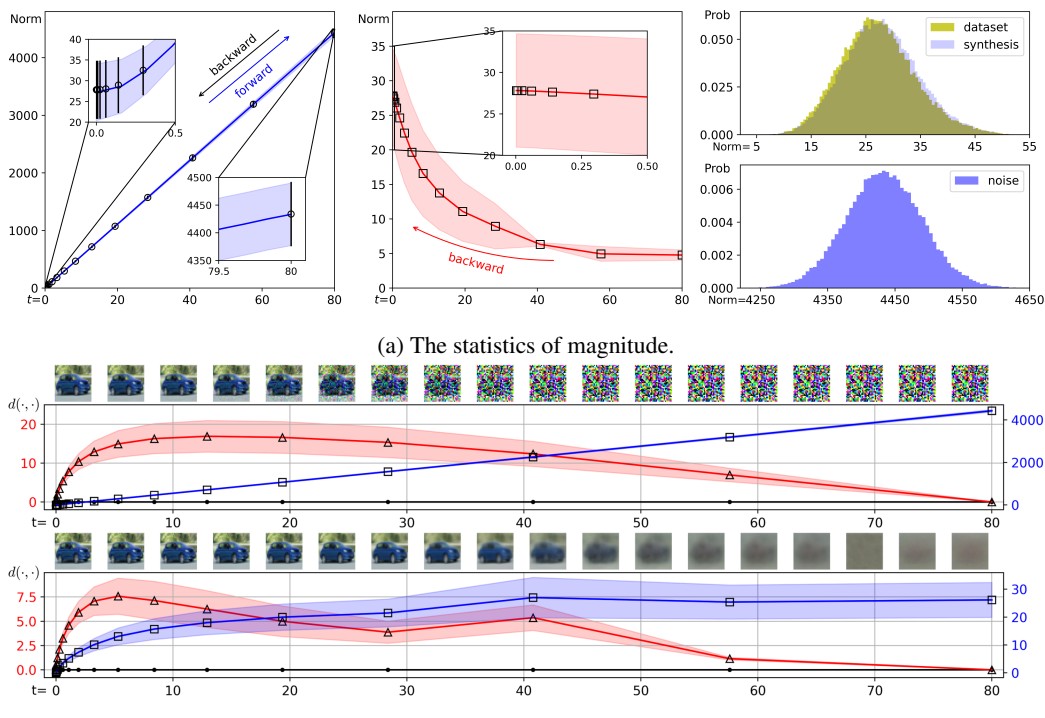

(a) The statistics of magnitude.

(b) Deviation in the sampling trajectory (top) and the denoising trajectory (bottom).

Figure 20: The unconditional generation with VE-SDE and NCSN++ architecture.

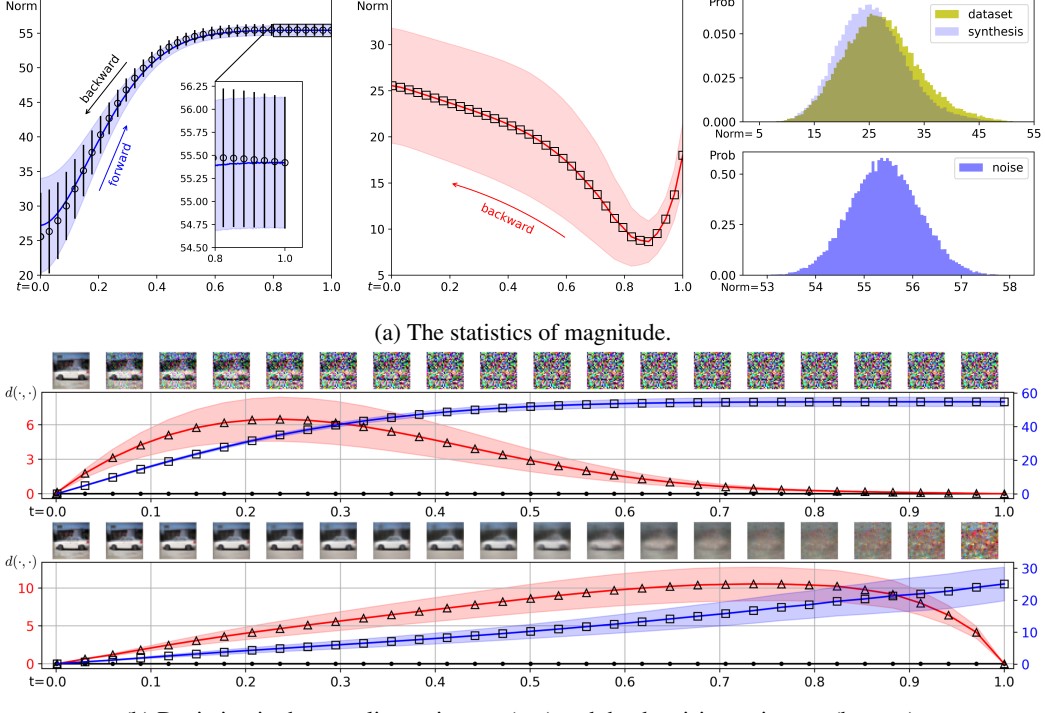

(a) The statistics of magnitude.

(b) Deviation in the sampling trajectory (top) and the denoising trajectory (bottom).

Figure 21: The unconditional generation with VP-SDE and DDPM++ architecture.

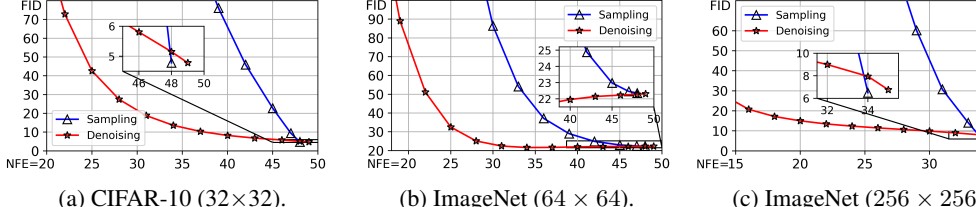

(a) CIFAR-10 (32×32).   (b) ImageNet (64 × 64).   (c) ImageNet (256 × 256).

Figure 22: The comparison of Fréchet Inception Distance (FID (Heusel et al., 2017), lower is better) *w.r.t.* the number of score function evaluations (NFEs). The sampling trajectory is simulated by the DPM-Solver sampler (Lu et al., 2022). The denoising trajectory still converges considerably faster than the sampling trajectory in this case.

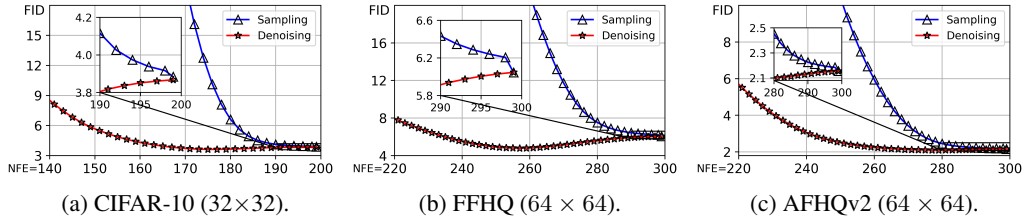

(a) CIFAR-10 (32×32).   (b) FFHQ (64 × 64).   (c) AFHQv2 (64 × 64).

Figure 23: The comparison of Fréchet Inception Distance (FID (Heusel et al., 2017), lower is better) *w.r.t.* the number of score function evaluations (NFEs). The sampling trajectory is simulated by second-order stochastic sampler (Karras et al., 2022). The denoising trajectory still converges considerably faster than the sampling trajectory in this case.

## F.5 MORE VISUAL RESULTS

- Figure 22 provides more comparison of the sampling trajectory *v.s.* the denoising trajectory on different datasets, with the default setting of DPM-Solver sampler[15] (Lu et al., 2022). For CIFAR-10, we adopt DPM-Solver-3 with $N = 18$, NFE = 48; for ImageNet-64, we adopt DPM-Solver-3 with $N = 18$, NFE = 48; for ImageNet-256 (with guidance), we adopt DPM-Solver-2 with $N = 18$, NFE = 34.

- Figure 23 provides more comparison of the sampling trajectory *v.s.* the denoising trajectory on different datasets, with the second-order stochastic sampler[16] (Karras et al., 2022).

- Figure 24 provides more comparison of the sampling trajectory *v.s.* the denoising trajectory on different datasets, with Heun's 2nd order method (Karras et al., 2022) for ODE-based sampling as the main submission. We calculate FIDs along the sampling trajectory and the denoising trajectory. Our proposed ODE-Jump sampling converges significantly faster than EDMs in terms of FID. More results are provided in Figures 26-28.

- Figures 25, 29 and 30 provide more comparison of the sampling trajectory *v.s.* the denoising trajectory on Stable Diffusion[17] (Rombach et al., 2022). We adopt the checkpoint "sd-v1-4.ckpt", and compute FID following the original paper, *i.e.*, comparing 30,000 synthetic samples ($256 \times 256$ resolution) with the validation set of MS-COCO-2014 dataset.

---

[15]https://github.com/LuChengTHU/dpm-solver
[16]https://github.com/NVlabs/edm
[17]https://github.com/CompVis/stable-diffusion/

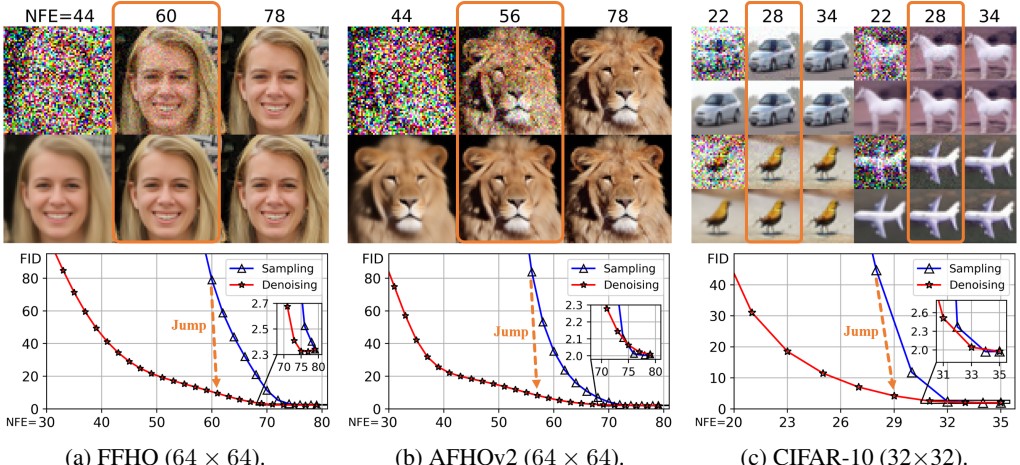

(a) FFHQ (64 × 64).          (b) AFHQv2 (64 × 64).          (c) CIFAR-10 (32×32).

Figure 24: The comparison of visual quality (top is sampling trajectory, bottom is denoising trajectory) and Fréchet Inception Distance (FID (Heusel et al., 2017), lower is better) *w.r.t.* the number of score function evaluations (NFEs). The sampling trajectory is simulated by Heun's 2$^{nd}$ order method (Karras et al., 2022) as the main submission. The denoising trajectory converges considerably faster than the sampling trajectory in terms of FID and visual quality. We thus employ ODE-Jump to obtain decent samples with a much less NFE.

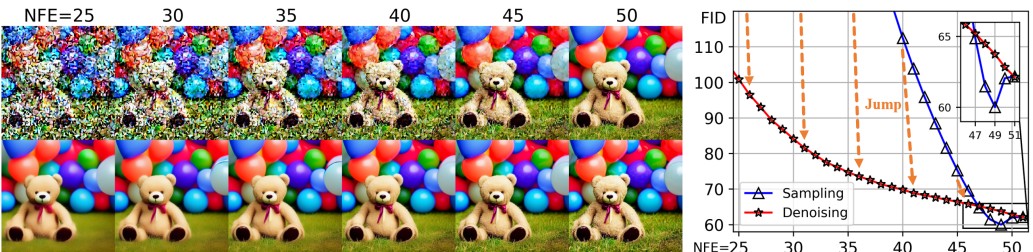

Figure 25: Stable Diffusion Results (Rombach et al., 2022). The comparison of visual quality (top is sampling trajectory, bottom is denoising trajectory) and Fréchet Inception Distance (FID (Heusel et al., 2017), lower is better) *w.r.t.* the number of score function evaluations (NFEs). The sampling trajectory is simulated by the default PLMS sampler with $N = 50$, NFE = 50. The denoising trajectory converges considerably faster than the sampling trajectory in terms of FID and visual quality. We thus employ ODE-Jump to obtain decent samples with a much less NFE.

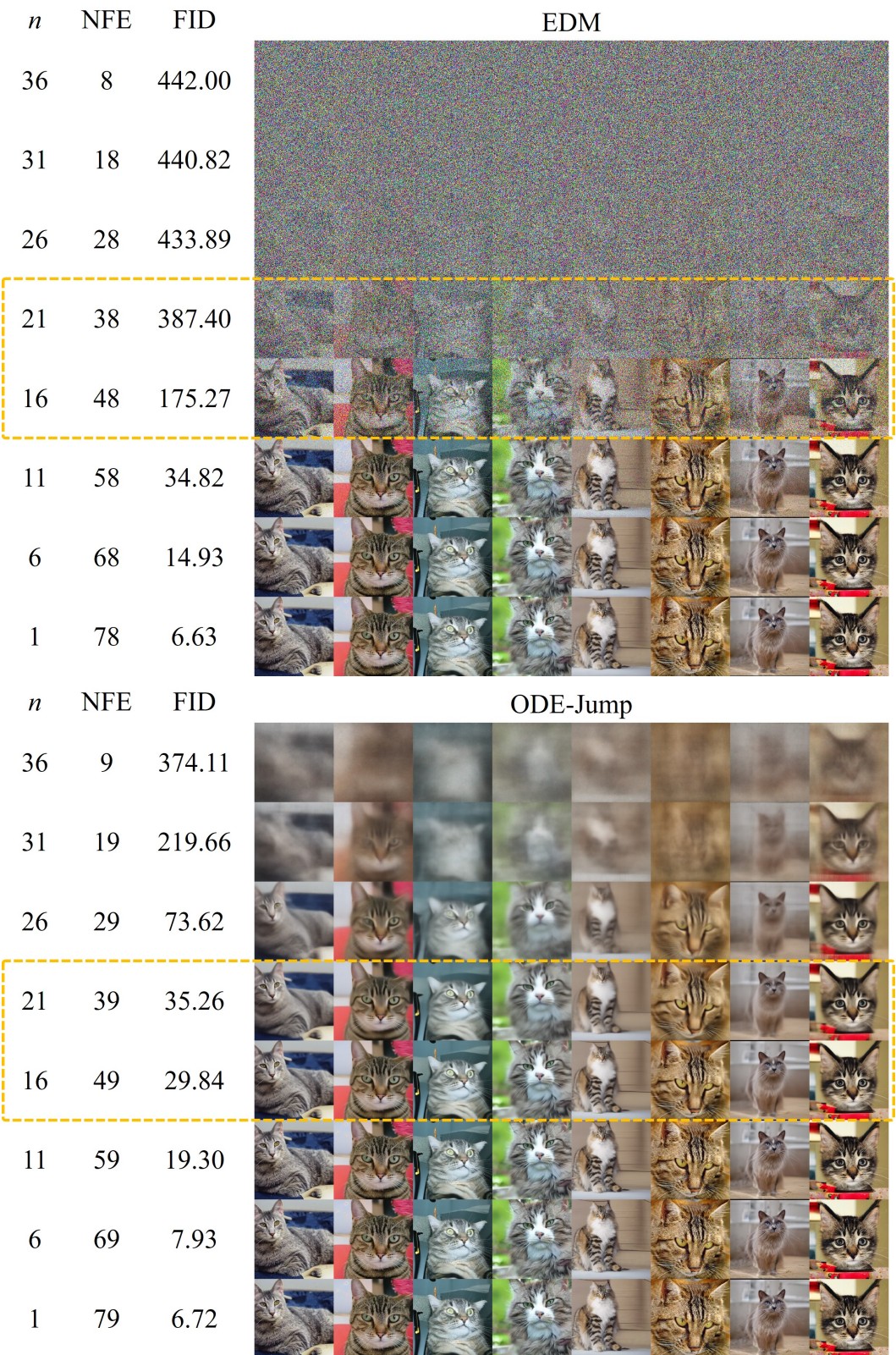

Figure 26: The visual comparison of EDM and ODE-Jump on LSUN Cat.

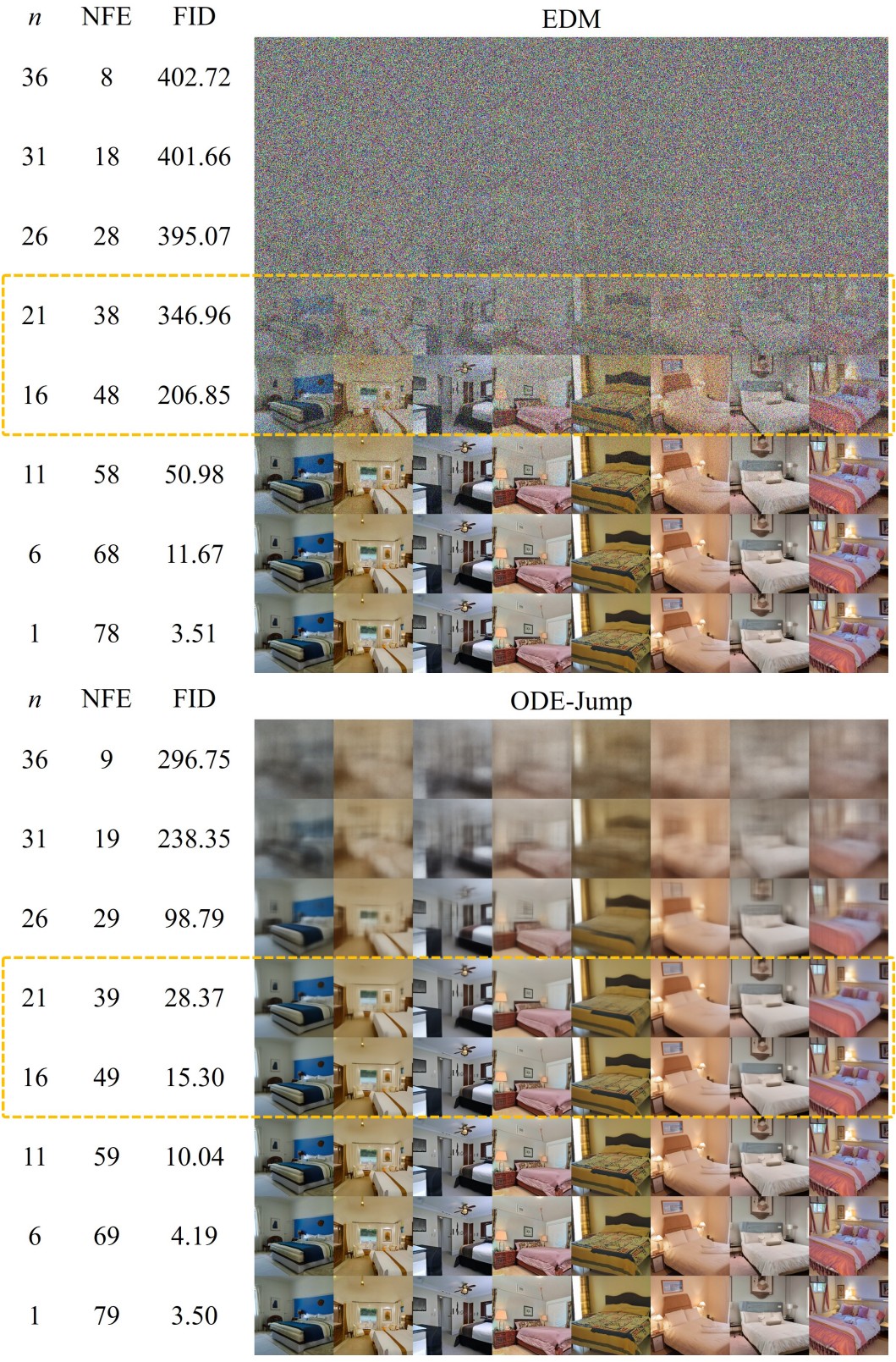

| n | NFE | FID | EDM |
|---|-----|-----|-----|
| 36 | 8 | 402.72 | |
| 31 | 18 | 401.66 | |
| 26 | 28 | 395.07 | |
| 21 | 38 | 346.96 | |
| 16 | 48 | 206.85 | |
| 11 | 58 | 50.98 | |
| 6 | 68 | 11.67 | |
| 1 | 78 | 3.51 | |

| n | NFE | FID | ODE-Jump |
|---|-----|-----|----------|
| 36 | 9 | 296.75 | |
| 31 | 19 | 238.35 | |
| 26 | 29 | 98.79 | |
| 21 | 39 | 28.37 | |
| 16 | 49 | 15.30 | |
| 11 | 59 | 10.04 | |
| 6 | 69 | 4.19 | |
| 1 | 79 | 3.50 | |

Figure 27: The visual comparison of EDM and ODE-Jump on LSUN Bedroom.

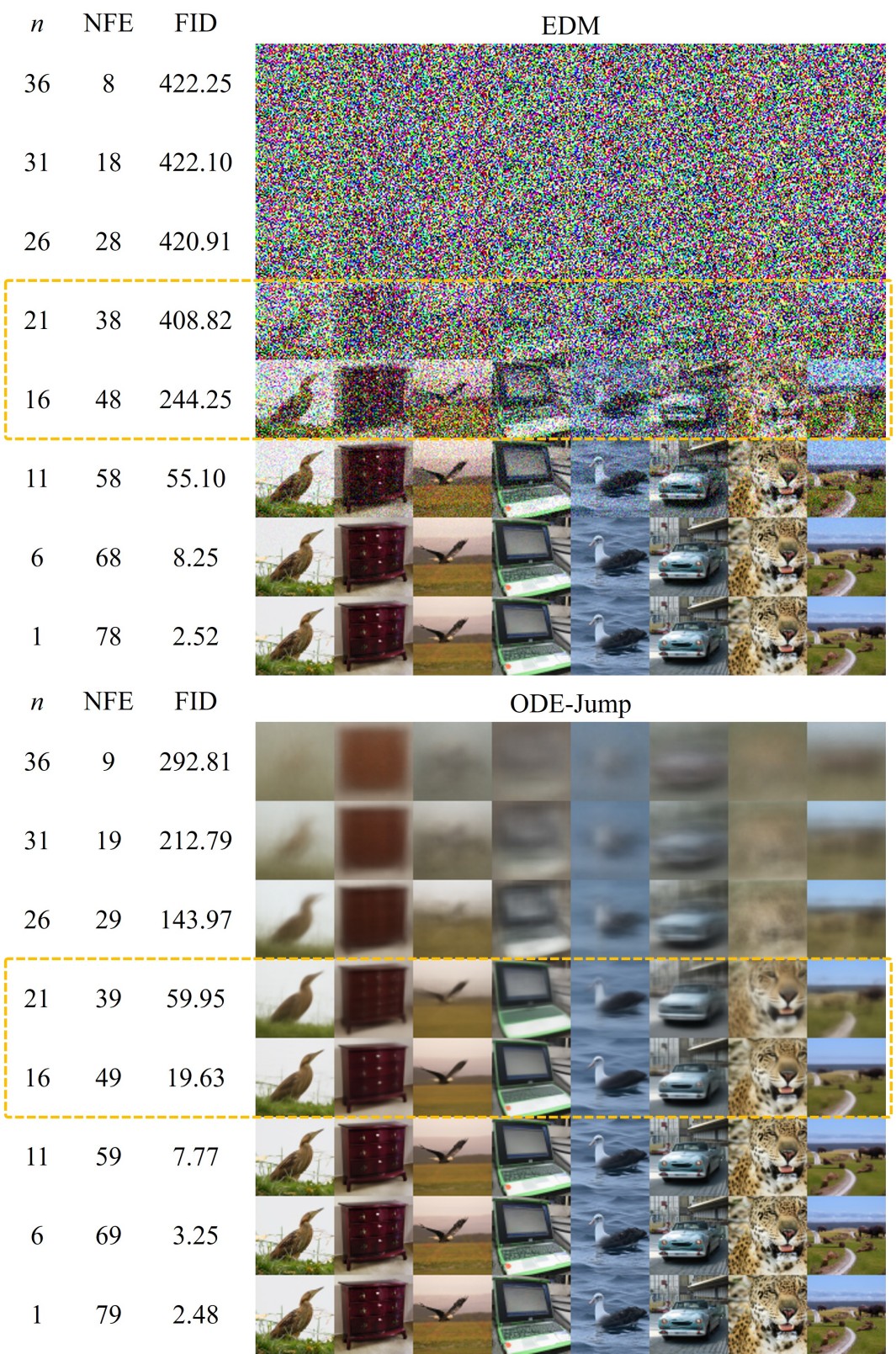

Figure 28: The visual comparison of EDM and ODE-Jump on ImageNet-64.

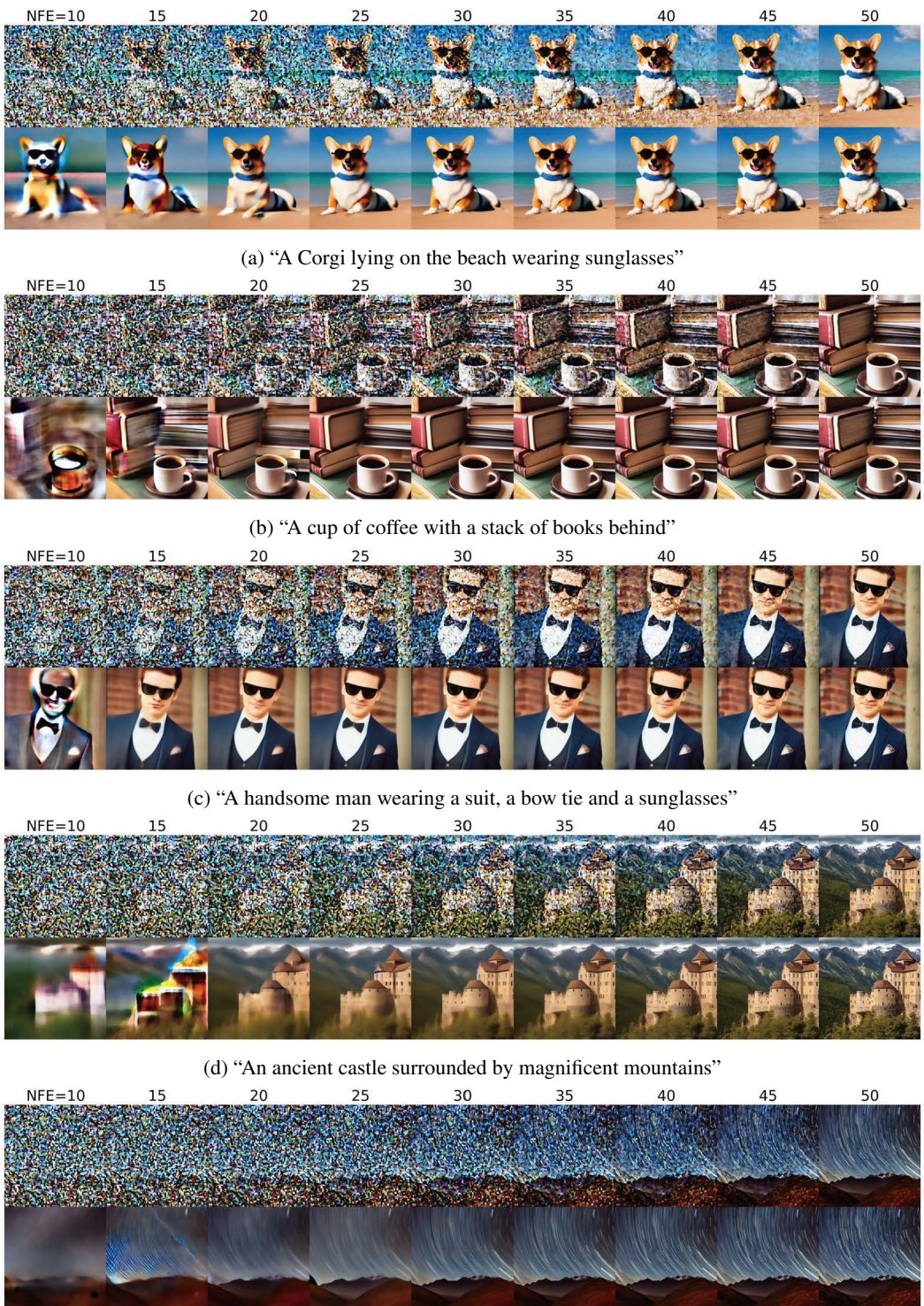

(a) "A Corgi lying on the beach wearing sunglasses"

(b) "A cup of coffee with a stack of books behind"

(c) "A handsome man wearing a suit, a bow tie and a sunglasses"

(d) "An ancient castle surrounded by magnificent mountains"

(e) "Long-exposure night photography of a starry sky over mountains, with light trails"

Figure 29: Stable Diffusion (PLMS sampler with $N = 50$, NFE $= 50$).

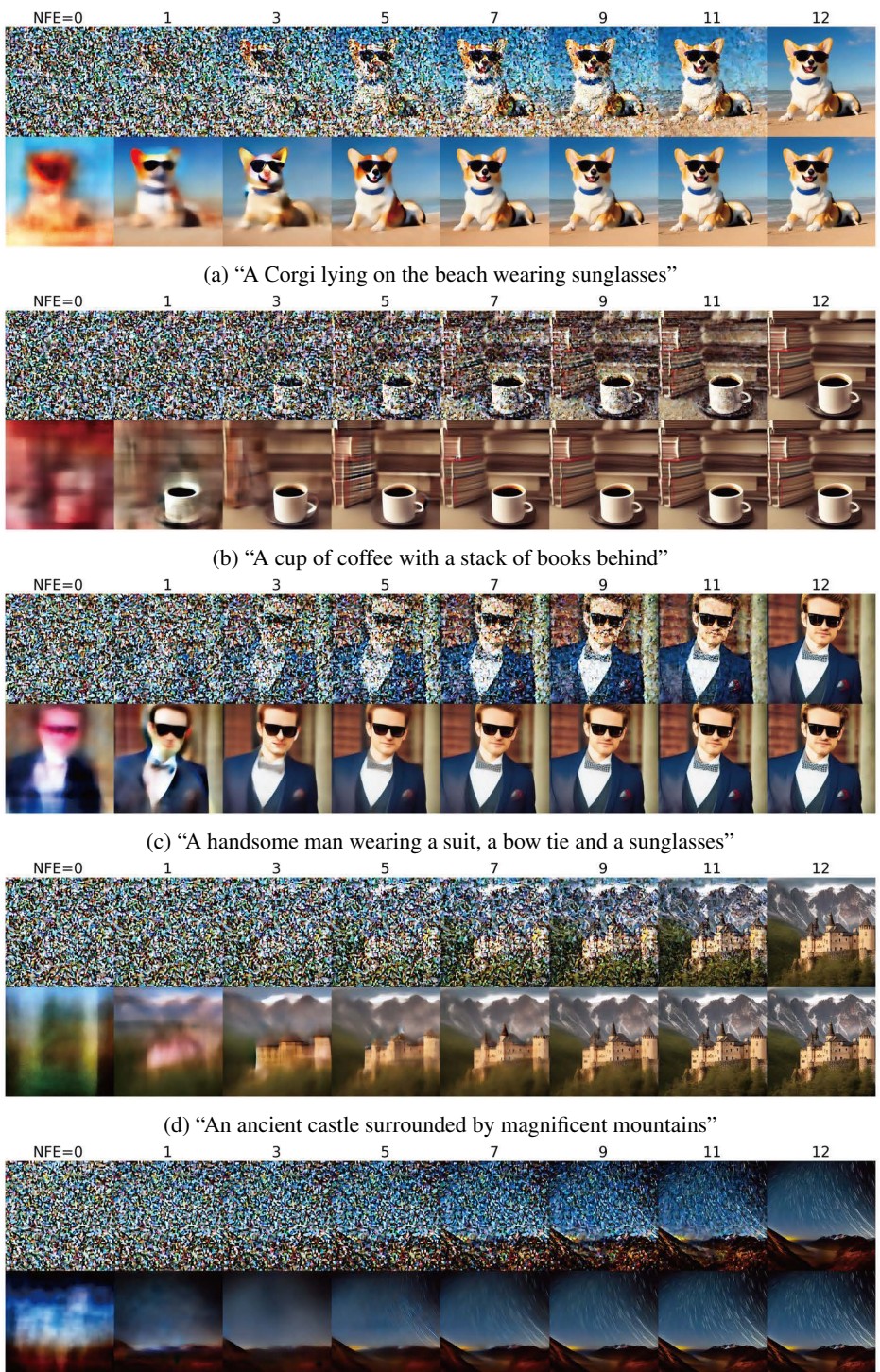

(a) "A Corgi lying on the beach wearing sunglasses"

(b) "A cup of coffee with a stack of books behind"

(c) "A handsome man wearing a suit, a bow tie and a sunglasses"

(d) "An ancient castle surrounded by magnificent mountains"

(e) "Long-exposure night photography of a starry sky over mountains, with light trails"

Figure 30: Stable Diffusion (DPM-Solver-2 with $N = 7$, NFE $= 12$).

