# OpenReview forum: "A Geometric Perspective on Diffusion Models"
_ICLR.cc/2024/Conference — ICLR 2024 Conference Withdrawn Submission_

### Official Review · Reviewer_C4jg · 2023-10-26

**Soundness:** 3 good
**Presentation:** 3 good
**Contribution:** 1 poor
**Rating:** 3
**Confidence:** 4

**Summary:**

This work proposes a geometrical analysis of the forward and backward dynamics associated to Variance Exploding-SDE diffusion models. In particular, authors compare the sampling and denoising trajectories, investigating their quasi-linearity and the fact that both move monotonically from their starting point to their final one.

The authors investigate how the output of the denoiser (obtained via reparametrization of the score) converges to a solution with low FID score much faster than the corresponding sampling trajectory. The authors proceed to discuss the relationship between higher order schemes and the denoising trajectory (Section 4) and the connection to mean shift (Section 5). Finally, the authors discuss the deviation of the ideal and parametric score, linking its behaviour to generalization.

**Strengths:**

* The paper's geometrical analysis of the forward and backward dynamics associated with Variance Exploding-SDE diffusion models is interesting and adds a new dimension to the understanding of these models. This type of analysis can be crucial for developing a deeper understanding of the underlying mechanisms of SDE-based models and their behavior during the sampling and denoising processes. Such insights can potentially inform the design of more efficient or accurate diffusion models in the future.

* Despite the focus on the VE-SDE case, the paper provides a detailed analysis of the sampling and denoising trajectories. The investigation into the quasi-linearity of these trajectories and their monotonic progression from start to finish is an interesting observation.

* Overall, the article is well written.

**Weaknesses:**

* One major concern is that the authors focus only on the VE case (although they mention that a similar analysis applies to a variance preserving SDE). While interesting, such limitation reduces generality of the claims. Some of the numerical observations might simply be artifacts of the considered SDE class, and do not provide information about diffusion models in general. Moreover, I find the different sections to be somehow disconnected; in certain cases some results are presented as original, while to the best of my understanding, they are restating known literature results. Finally, the connection to a “geometrical perspective” is very weak.

* Recall Observation 1: “The sampling trajectory is almost straight while the denoising trajectory is bent”. As explicitly mentioned in Proposition 2, the norm of the corrupted data will be much greater than the clean data. On the other hand, the denoising trajectory will have (by construction) a roughly constant norm at each step. Then observation 1 is almost trivial. Is it still valid for other SDEs? If not, what information is the proposition offer to the reader? Similar considerations apply for Observation 2. Also, I would advise the authors to rescale Fig 2b (maybe logarithmic y axes), since in the “region of interest”(small diffusion steps) it is very difficult to compare top and bottom subplots.

* Observation 3 is equivalent to the following rephrasing: “the output of a denoiser has better FID score than its input”. This is trivially true, and once again provides very limited insight into the geometry of diffusions. One interesting technique is related to the computation of the FID score using the JUMP scheme, but it seems that the technique provides limited benefit for values of FID which are competitive.

* Section 4 discusses how second order schemes use in their numerical implementation approximations of the second order time derivative of the sampling process. This is exactly how second order numerical
schemes are derived. Then, what is the take home message of this Section?

* Proposition 4 is a known result from [Karras 2022], in particular Eq. (57) in the Appendix. The authors should explicitly mention this. I would also suggest to rephrase “once a diffusion model has converged to the optimum...”. At first I was confused about whether the authors were referring to convergence to the final
sampling point or convergence of the parameters of the score network (I assume the authors are referring to the latter).

* The content of Section 6 concerns the study of score deviation from its ideal value. Also in this case, the authors do not to compare their results with the work by Karras, 2022, where it is stated (see their
Section 5, page 9):
“Inspecting the per-$\sigma$ loss after training (blue and orange curves) reveals that a significant reduction is possible only at intermediate noise levels; at very low levels, it is both difficult and irrelevant to discern the vanishingly small noise component, whereas at high levels the training targets are always dissimilar from the correct answer that approaches dataset average”
How is this different from the claim in your article: “ the deviation between the learned denoising output  and its optimal counterpart behaves differently in three successive regions...”?

* Finally, while the content of Section 6 is technically correct, I fail to see the geometrical perspective in the findings (is the l2 distance the only geometrical metric to consider?)

**Questions:**

* Given that the study focuses exclusively on Variance Exploding-SDE (VE-SDE) models, how do the authors anticipate their findings would generalize to other types of SDEs, such as Variance Preserving-SDEs (VP-SDEs)? Could the geometrical insights and behavior of trajectories observed in this study be applicable or significantly different for other SDE formulations?

* The review on the weaknesses above points out that the connection to a "geometrical perspectives" seems weak in some sections. Could the authors elaborate on how they define and utilize geometrical analysis within the context of this study, particularly how it pertains to understanding the dynamics of diffusion models?

* Regarding Observation 3, where it's noted that the output of a denoiser generally has a better FID score than its input, could the authors discuss any unexpected or non-trivial implications of this finding, particularly in how it might influence the design or optimization of future diffusion models?

* Proposition 4 appears to overlap significantly with findings from [Karras 2022]. Could the authors clarify what distinct contribution this proposition makes beyond the existing literature? Additionally, how do their findings in Section 6 diverge from or confirm the observations made in [Karras 2022] regarding score deviation at various noise levels?

* Could you please help in clarifying what are the novel contributions of Section 4, with respect to the discussion on second order schemes? Could the authors highlight how their analysis of these schemes contributes new understanding or practical advancements in the field of diffusion models?

---

### Official Review · Reviewer_dBDN · 2023-10-29

**Soundness:** 1 poor
**Presentation:** 2 fair
**Contribution:** 1 poor
**Rating:** 3
**Confidence:** 5

**Summary:**

The paper aims to study the variance-exploding diffusion models used for generative modeling. It contains some propositions, lemmas, and a theorem stating some theoretical properties of the diffusion models. These results characterize the behavior of the denoising and sampling trajectories. They are often written in a sloppy manner and most of them, in my opinion, concern some simple facts that do not need to be stated as propositions.

The paper also contains some observations, based on empirical experiments on image datasets that are often used in the literature on generative modeling. These observations are mostly some simple claims that are not poorly justified. In my opinion, their practical usefulness is limited. Some precise remarks are provided below, but let me simply quote here Observation 4 "The learned score is well-matched to the optimal score in the large-noise region, otherwise they may diverge or almost coincide depending on different regions." Basically, this observation tells us that in the small-noise region, which is the main region of interest, one cannot say much about the fit of the optimal score with the learned score.

**Strengths:**

1) The paper deals with an interesting problem, that of generative modeling using exploding-variance diffusion models.
2) To the best of my knowledge, Theorem 1 is new.
3) Experimental results are of some interest.

**Weaknesses:**

1) Mathematical writing is sloppy.
2) The observations are not sufficiently well justified, and their impact is limited.
3) Some of the mathematical results are trivial.

More specific remarks:

Section 2 :

- There is an ambiguity between distribution and density function. Given the formula of the score function used, for instance, in (2), $p_t$ denotes the probability density function with respect to the Lebesgue measure. Then, it is written that $p_0 = p_d$ is the empirical data distribution. This is unclear to me. The empirical data distribution does not admit a density wrt to the Lebesgue measure. I guess $p_d$ is not the empirical data distribution, but the density of the “theoretical” distribution of a data point.

- It is worth defining the precise meanings of $p_d$ and $p_n$ in the beginning of Section 2
- Eq 1 & 2, $dx$ should be replaced by $dx_t$
- Eq 3: this part is unclear for several reasons. First, it is not defined what is $\theta$, what is $\sigma$ and what is $r$? Second, the least squares problem should necessarily involve a minimization; it is not specified wrt to which quantity the minimization should be done. Third, what is the output of the least-squares problem and how it is related to the score?
- Proposition 1 can hardly be considered as a mathematical statement. Is it possible to formulate it in the conventional forme: let $r_{\theta}$  be … and $\tilde x$ be the output of a one-step Euler scheme …, then … ? Without exaggeration, with the current formulation I do not understand the claim of the proposition.
- From a mathematical point of view, the first claim of Prop 2 is not well formulated. Indeed, the quantity on the left-hand side is random, which means that the meaning of the convergence hidden in big-O notation should be made clear. Regarding the proof of this proposition presented in the appendix,
  * Eq 44 has no mathematical meaning. One cannot write $a = b \pm c$.
  * Eq 45 is wrong
I believe what the authors try to do in this proposition can be done in a clean way
using Talagrand’s Gaussian concentration inequality.

Section 3 :
- Observation 1 is poorly justified. The fact that the distance between the 10 points of the trajectory and the straight line connecting the first point to the last point of the set is smaller than 20 does not necessarily imply that the continuous-time path these points are sampled from is a line. It might very well be a spiral around a line or any other curve.
There is a well known method in statistical data analysis for measuring how close a point cloud is to a line. It is called PCA and suggests to look at the spectrum of the correlation matrix. It would be more relevant to justify the closeness to a line by such a method, based on more than a dozen of points.
- Observation 2: it is not quite clear to me what the authors call “move monotonically”. Since the justification of observation 2 seems to be the blue curve of fig 2 (b), I guess the meaning of monotonicity here is to be understood as the monotonicity of the function that maps t to the distance between z(t) and z(0), where z is either the x or r.
- Observation 3, and Fig 3: I am not sure that the denoising trajectory converges faster than the sampling trajectory does. The speed of convergence is the derivative of the curve (the slope), which seems to be stronger for the sampling trajectory than for the denoising trajectory.

**Questions:**

I have no questions for the authors. My suggestions are listed above.

---

### Official Review · Reviewer_SvFU · 2023-10-31

**Soundness:** 3 good
**Presentation:** 2 fair
**Contribution:** 2 fair
**Rating:** 3
**Confidence:** 2

**Summary:**

This paper examined the geometric properties of the sampling trajectory and the denoising trajectory, along an ODE-based sampling process, EDM. The observed properties are summarized as following:

1. The sampling trajectory is almost straight while the denoising trajectory is bent.

2. Both trajectories monotonically move from the initial points to final points in expectation.

3. The denoising trajectory converges faster than the sampling trajectory in terms of visual quality, FID and sample likelihood.

4. The learned score is well-matched to the optimal score in the large-noise region, otherwise they may diverge or almost coincide depending on different regions.

Based on the observations, the authors proposed that

1. an ODE-jump sampler, which jumps from the sampling trajectory to the denoising trajectory in the last step, can achieve FID improvement/reduce NFEs.

2. a series of second-order sampler can be reduced to a second-order Taylor polynomial approximation of the sampling trajectory and various finite differences of derivative of the denoising trajectory.

At the end of the paper, the optimal denoising output was analyzed and related to annealed mean shift.

**Strengths:**

1. This paper combines lots of analytical and experimental evidences to support the geometric understanding of the sampling trajectory and the denoising trajectory in EDM.

**Weaknesses:**

1. Lack of novelty. Most of the results/techniques have been observed/used in other papers.

**Questions:**

1. How are section 4 and 5 related to the geometric properties of the two trajectories? I don't quite understand why it is essential to include  Section 4 in the paper.

2. For the ODE-jump solver, can we actually prove the improvement theoretically?

---

### Official Review · Reviewer_fQGm · 2023-11-01

**Soundness:** 4 excellent
**Presentation:** 4 excellent
**Contribution:** 1 poor
**Rating:** 5
**Confidence:** 3

**Summary:**

This work explores ODE-based sampling from variance exploding diffusion models from a geometric perspective. In particular, the relationship between the sampling trajectory and the denoising trajectory is investigated. Several small propositions and observations are made which link existing fast ODE solvers and ODE-based sampling to the mean-shift algorithm.

**Strengths:**

- Very clear exposition and intuitive geometric explanation for some interesting observations on ODE-based sampling of VE diffusion models
- Simple yet effective theory linking mean-shift algorithms to diffusion models which I have not seen before.
- Theory is supported by empirical evidence in large-scale diffusion models.

**Weaknesses:**

- My main issue with this work is that it is unclear exactly how to use these theoretical insights and observations to improve diffusion models and diffusion model sampling. While it is interesting to link mean-shift algorithms to diffusion models it is not shown how this link leads to improved sampling. Similarly, while all the fast ODE solvers are shown to be related, it is unclear how to use this for better sampling.
- Theory is all quite simple (though useful), and is not particularly novel.
- I have seen the jump trick used in practice fairly often, and do not think this is a particularly novel insight. For example in RFDiffusion this trick is used [1]. Though I doubt this is the origin of this trick.

Overall I lean towards rejection, but could be convinced by a small amount of actionable insight, given the clear exposition and intuitive geometric figures. However, I note I do not have full knowledge of the novelty of this with regard to related work in this space.

[1] Joseph L. Watson, David Juergens, Nathaniel R. Bennett, Brian L. Trippe, Jason Yim, Helen E. Eisenach, Woody Ahern, Andrew J. Borst, Robert J. Ragotte, Lukas F. Milles, Basile I. M. Wicky, Nikita Hanikel, Samuel J. Pellock, Alexis Courbet, William Sheffler, Jue Wang, Preetham Venkatesh, Isaac Sappington, Susana Vázquez Torres, Anna Lauko, Valentin De Bortoli, Emile Mathieu, Regina Barzilay, Tommi S. Jaakkola, Frank DiMaio, Minkyung Baek, David Baker. Broadly applicable and accurate protein design by integrating structure prediction networks and diffusion generative models. Science 2023.

**Questions:**

Comments:

Prop 2 could use an “and” before the limit.

> We mostly take unconditional generation on CIFAR-10 as an example to demonstrate our observations.

Is this true? I don’t see this from the figures.

---

### Author Response · Authors · 2023-11-21
**A few things to clarify (especially for future readers)**

We are shocked that this paper received UNANIMOUS REJECT Ratings from ICLR, given than it previously received (Five) UNANIMOUS ACCEPT Ratings from NeurIPS.

We briefly clarify a few points below:

- **For Reviewer fQGm:** We consider our new perspective on diffusion models and theoretical insights as the main contribution of this paper, rather than a new SOTA. One take-home message for better sampling: Try to leverage the implicitly denoising trajectory with a fast convergence speed. Some successful distillation-based samplers have made initial attempts in a heuristic way, without being aware of this fact.
- **For Reviewer SvFU:**
1. Please provide the exact references instead of using a phrase “lack of novelty”.
2. To understand the implicit denoising trajectory, we certainly need to discuss its differential equation and its theoretical relationship with the commonly used sampling trajectory.
3. We have provided a theoretical guarantee (Theorem 1) and its proof (Appendix C.1) in our submission.
- **Reviewer dBDN:**
1. We follow the notations and derivations of two influential papers “Score-Based Generative Modeling Through Stochastic Differential Equations”, “Elucidating the Design Space of Diffusion-Based Generative Models”, and the textbook “High-Dimensional Probability” written by Roman Vershynin. Anyone familiar with diffusion models should feel comfortable with our mathematical expressions.
2. The observation based on PCA aligns with our current results and does not alter our conclusion.
- **Reviewer C4jg:**
 1.  Your main concern has already been theoretically addressed in Section 7 and Appendix A.2 of our submission.

1. We aim to provide a geometric picture for an intuitive understanding of the diffusion sampling. To achieve this goal, we need to reveal the differential equations of two trajectories as well as their relationship (Section 4), provide a theoretical support for our observations (Sections 3 and 5), and explain how the actual sampling trajectory deviates from the optimum (Section 6).
2. “the denoising trajectory will have (by construction) a roughly constant norm at each step”. However, your intuition is not true according to experiments.
3. “I would advise the authors to rescale Fig 2b”. This has been provided in Figure 13 of our submission.
4. “the output of a denoiser has better FID score than its input. This is trivially true”. Again, your intuition is not true in general case.
5. Proposition 4 is a standard textbook result for least squares estimation (though our derivation differs from [Karras 2022]). Obviously, we do not consider Prop. 4 as our contribution but include it for better comparison with mean shift.
6. If you had a good understanding of our paper or [Karras 2022], you would realize that our Section 6 is fundamentally different from their Section 5.

Overall, we appreciate the valuable part of reviews, but not those ridiculous comments. We decide to withdraw this submission without further response.